# Balanced partial entanglement and mixed state correlations

Hugo A. Camargo[1,2], Pratik Nandy[3], Qiang Wen[4,5*] and Haocheng Zhong[4]

1 Max-Planck-Institut für Gravitationsphysik ,
Am Mühlenberg 1, 14476 Potsdam-Golm, Germany
2 Dahlem Center for Complex Quantum Systems, Freie Universität Berlin,
Arnimallee 14, 14195 Berlin, Germany
3 Centre for High Energy Physics, Indian Institute of Science,
C.V. Raman Avenue, Bangalore 560012, India
4 Shing-Tung Yau Center of Southeast University, Nanjing 210096, China
5 School of Mathematics, Southeast University, Nanjing 211189, China

★ wenqiang@seu.edu.cn

## Abstract

Recently in Ref. [1], one of the authors introduced the balanced partial entanglement (BPE), which has been proposed to be dual to the entanglement wedge cross-section (EWCS). In this paper, we explicitly demonstrate that the BPE could be considered as a proper measure of the total intrinsic correlation between two subsystems in a mixed state. The total correlation includes certain crossing correlations, which are minimized by particular balance conditions. By constructing a class of purifications from Euclidean path-integrals, we find that the balanced crossing correlations show universality and can be considered as the generalization of the Markov gap for the canonical purification. We also test the relation between the BPE and the EWCS in three-dimensional asymptotically flat holography. We find that the balanced crossing correlation vanishes for the field theory invariant under $BMS_3$ symmetry (BMSFT) and dual to the Einstein gravity, indicating the possibility of a perfect Markov recovery. We further elucidate these crossing correlations as a signature of tripartite entanglement and explain their interpretation in both AdS and non-AdS holography.

 Check for updates

# 1  Introduction

The structure of quantum entanglement in quantum systems plays a fundamental role in understanding the quantum information-theoretic nature of quantum gravity. The quantum nature of entanglement is captured by the entanglement entropy, which has been extensively studied in quantum field theories and many-body systems [2]. In the context of AdS/CFT correspondence [3–5], the entanglement entropy in the field theory of a region $A$ in the boundary has a dual description in terms of the area of a minimal surface $\mathcal{E}_A$ in the dual gravity side and goes by the name of Ryu-Takayanagi formula [6–8]

$$S_A = \frac{\text{Area}\,(\mathcal{E}_A)}{4G_N}\,. \tag{1}$$

The entanglement entropy is a good measure of quantum entanglement between a subsystem and its complement when the boundary state is pure. However, for mixed states, the entanglement entropy is not a proper measure to capture the intrinsic correlation between the subsystems [9, 10]. Recently, the study of said correlation in mixed states has attracted considerable attention. Several quantities have been defined to measure the mixed state correlations, like the mutual information, the (logarithmic) entanglement negativity [11–18], and the entanglement of purification (EoP) [19–21]. From the dual gravity side, this correlation is supposed to be captured by a special geometric quantity, known as the entanglement wedge cross-section (EWCS). It is a natural candidate to measure the intrinsic correlation between the two subsystems in a mixed state with a gravity dual. Moreover, in order to understand the EWCS better, new information-theoretic quantities have been proposed by the high-energy physics community, such as the reflected entropy [22], the "odd entropy" [23], the "differential purification" [24], entanglement distillation [25, 26] and the balanced partial entanglement (BPE) [1, 27, 28]. See [29–43] for some recent explorations on the study of the purification and the EWCS.

In Ref. [1], based on the concept of the partial entanglement entropy (PEE) and its holographic picture, it was observed that the PEE satisfying certain balance conditions could be

considered as the area of the EWCS in the dual gravity picture. This specific PEE is called the *balanced partial entanglement entropy* (BPE). In principle, the definition of BPE is more general, and it can be defined for any purification, including the non-holographic ones. For the specific case of the canonical purification, the BPE reduces to half of the reflected entropy, suggesting BPE as a more generic measure than the reflected entropy. It has been observed BPE obeys the entropy relations that are satisfied by the EoP and the EWCS.

In this paper, we take a step towards understanding why the BPE could be a proper measurement of the total intrinsic correlation between the subsystems $A$, and $B$ is a mixed state $\rho_{AB}$. For such a mixed state, it is useful to consider an auxiliary system $A'B'$, which together with $AB$, forms a pure state. This pure state is called the purification of the mixed state and is highly non-unique. A meaningful way to explore the correlation structure in a mixed state is to study the entanglement entropy $S_{AA'}$ in the purifications. It is important to note that some of the correlations, for example, the correlation inside $A'B'$, contribute to $S_{AA'}$ but not to the intrinsic correlation inside $AB$. We need to either minimize those correlations or try to exclude them from $S_{AA'}$ in a proper way. This is how the EoP and the reflected entropy are defined. In the following, we briefly introduce the EWCS, the EoP, and the reflected entropy.

*Entanglement wedge cross section (EWCS)*: Let us consider a region in the boundary field theory with a partition $AB \equiv A \cup B$, described by a reduced density matrix $\rho_{AB}$. Its holographic dual is a bulk region known as the entanglement wedge [44–46]. For a static time slice, the entanglement wedge is the region enclosed by the boundary subregion and the corresponding minimal surface. This allows us to define the EWCS $\Sigma_{AB}$ as the minimal area cross-section separating the regions $A$ and $B$.

*Entanglement of purification (EoP)*: Let $|\Psi\rangle \in \mathcal{H}_{AA'} \otimes \mathcal{H}_{BB'}$ be any purification of $\rho_{AB}$. One defines the EoP $E_p(A:B)$ [19] as

$$E_p(A:B) = \min_{|\Psi\rangle, A'} S_{AA'}, \tag{2}$$

where we take the minimization over all purifications of $AB$ and over all the possible partitions of $A'B'$. The EoP is then given by the minimal value of $S_{AA'}$. The minimization procedure in some sense excludes the contribution from the intrinsic correlation between $A'$ and $B'$ to $S_{AA'}$, hence could be a valid measure of correlation between $A$ and $B$.

*Reflected entropy*: Consider a bipartite system $AB$ associated with the Hilbert space $\mathcal{H}_{AB}$ and an orthonormal basis $\{|\Psi_i\rangle\}$ of $\mathcal{H}_{AB}$. In general, any mixed state can be written as

$$\rho_{AB} = \sum_i p_i |\Psi_i\rangle \langle \Psi_i|. \tag{3}$$

To define the reflected entropy, we need to introduce an auxiliary system $A'B'$, whose Hilbert space is the identical copy of the original Hilbert space. One then defines the canonical purification as

$$|\sqrt{\rho_{AB}}\rangle = \sum_i \sqrt{p_i} |\Psi_i\rangle_{AB} |\Psi_i\rangle^*_{A'B'}. \tag{4}$$

Here $\{|\Psi_i\rangle^*\}$ is an orthonormal basis of the Hilbert space $\mathcal{H}_{A'B'} = \mathcal{H}_{AB}$. The definition of reflected entropy is invoked through the definition the entanglement entropy for $AA'$ as

$$S_R(A:B) = S_{AA'} = -\text{Tr}\rho_{AA'} \log \rho_{AA'}, \tag{5}$$

where the mixed state $\rho_{AA'} = \text{Tr}_{BB'} |\sqrt{\rho_{AB}}\rangle\langle\sqrt{\rho_{AB}}|$ is obtained by tracing out the degrees of freedom in $BB'$. Since the complement $A'B'$ is just a copy of $AB$, the entanglement between $AA'$ and $BB'$ could be the double of the intrinsic total correlation in $AB$. Thus the total intrinsic correlation between $A$ and $B$ is captured by half of the reflected entropy.

Both the EoP and the reflected entropy are lower bounded by half of the mutual information $I(A : B)/2$, which we consider as a direct correlation between $A$ and $B$. They also get contributions from the "crossing correlations", including the correlations between $A$ and $B'$ and between $B$ and $A'$, which have not been explicitly studied before. We will study all these correlations in the context of the PEE. Hence the crossing correlations are also called the crossing PEEs, which we argue to be a generalized version of the Markov gap. We show that the crossing PEEs are minimized under some balance conditions, and they are independent of any purification. More interestingly, in $(1 + 1)$-dimensional CFTs, the crossing PEE is universal. The way BPE captures the total intrinsic correlation is similar to the reflected entropy, and, in fact, the partition of $A'B'$ in the canonical purification automatically satisfies the balance conditions, as we will see later sections.

The definition of the BPE can be applied to generic theories. Its correspondence to the EWCS should also go beyond AdS/CFT. The Markov gap or the crossing PEEs computed in AdS gravity is non-vanishing, and the non-vanishing Markov gap demands the existence of an approximate Markov recovery process. To understand the BPE and the Markov gap in general, we consider the duality between BPE and EWCS in flat holography. We explicitly compute the crossing PEEs for the field theory invariant under the $BMS_3$ symmetries (BMSFT), which is dual to the 3-dimensional flat space. The crossing PEE in the field theory dual to Einstein gravity identically vanishes; thus, it reflects the existence of a perfect Markov recovery process in BMSFT.

The structure of the paper is organized as follows. In section 2, we will briefly review the concept of PEE and the balanced PEE. Then in section 3 we construct a class of purifications for $\rho_{AB}$ from path-integral optimization and calculate the BPE. This calculation shows that the BPE is independent of this class of purifications. In section 4, we focus on the details of the balance conditions. We show that when the balance conditions are satisfied, the crossing PEEs reach their minimal value; hence, the BPE reasonably measures the total correlation in the mixed state. We also show that the crossing PEEs can be considered as a generalized version of the so-called Markov gap. In section 5, we discuss the details of the Markov recovery process and calculate the BPE for BMSFT. We show that the BPE coincides with the EWCS, and as a result, the crossing PEE vanishes for the BMSFT dual to the Einstein gravity. We summarize our results and give possible future directions in section 6.

## 2 The balanced partial entanglement

### 2.1 The partial entanglement entropy

We first introduce the concept of the *entanglement contour* [47]. It is a local measure of entanglement capturing the contribution coming from each degree of freedom inside a region $A$ to the entanglement entropy $S_A$. It is a function denoted by $s_A(x)$, where $x \in A$. Note that the function is non-local since it depends on the region $A$. This paper focuses on two-dimensional systems; hence, $x$ characterizes the spatial direction. We recover the entanglement entropy by collecting the contributions from all the sites inside $A$, hence $S_A = \int_A s_A(x)\,\mathrm{d}x$. It is sometimes more useful to study a quasi-local measure of entanglement, i.e., the so called *partial entanglement entropy* (PEE) $s_A(A_i)$. Instead of capturing the contribution from each degree of freedom, one is interested to consider the contribution from a subset of region $A_i \subset A$. One defines

$$s_A(A_i) = \int_{A_i} s_A(x)\,\mathrm{d}x\,. \tag{6}$$

The PEE $s_A(A_i)$ captures certain type of the correlation between the subregion $A_i$ and the system $\bar{A}$ that purifies $A$. Since the correlation is mutual, similar to the mutual information (MI), one can also denote the PEE as

$$s_A(A_i) \equiv \mathcal{I}(A_i, \bar{A}) \equiv \mathcal{I}_{A_i \bar{A}}. \tag{7}$$

One should be careful not to confuse the PEE $\mathcal{I}$ with the MI, $I(A : B) = S_A + S_B - S_{AB}$. We interchangeably use the notation between $s_A(A_i)$ and $\mathcal{I}(A_i, \bar{A})$.

The PEE should respect certain physical requirements [28,47]. For self consistency, we list them in the following:

1. *Additivity*: For any two spacelike-separated regions $B$ and $C$ such that $B \cap C = \emptyset$, the additivity says $\mathcal{I}(A, B \cup C) = \mathcal{I}(A, B) + \mathcal{I}(A, C)$.

2. *Unitary invariance*: $\mathcal{I}(A, B)$ should be invariant under any local unitary transformation inside the regions $A$ and $B$.

3. *Symmetry transformation*: For any symmetry transformation $\mathcal{T}$ such that $\mathcal{T}A = A'$ and $\mathcal{T}B = B'$, the PEEs should remain invariant i.e., $\mathcal{I}(A, B) = \mathcal{I}(A', B')$.

4. *Normalization*: $\mathcal{I}(A, B)|_{B \to \bar{A}} \to S_A$.

5. *Positivity*: $\mathcal{I}(A, B) \geq 0$.

6. *Upper bound*: $\mathcal{I}(A, B) \leq \min\{S_A, S_B\}$.

7. *The permutation symmetry between A and B*: $\mathcal{I}(A, B) = \mathcal{I}(B, A)$.

So far, there are several proposals[1] for the PEE that satisfy the above 7 requirements. In this paper, we will mainly use the *additive linear combination* (ALC) proposal [27,61] in two-dimensional field theories. It was shown in [27,28,58,61] that the ALC proposal satisfies all the above-mentioned properties.

- *The ALC proposal*:

  Consider a boundary region $A$. Suppose that it can be partitioned into three non-overlapping subregions $A = \alpha_L \cup \alpha \cup \alpha_R$, where $\alpha$ is some subregion inside $A$ and $\alpha_L$ ($\alpha_R$) denotes the regions left (right) to it. On this configuration, the claim of the *ALC proposal* is the following:

$$s_A(\alpha) = \mathcal{I}(\alpha, \bar{A}) = \frac{1}{2}\left(S_{\alpha_L \cup \alpha} + S_{\alpha \cup \alpha_R} - S_{\alpha_L} - S_{\alpha_R}\right). \tag{8}$$

The *ALC proposal* is supposed to be general and applicable for any theories. However, as we have stated before, a specific configuration and order between the subsets are required inside $A$. This order is essential for the PEE to satisfy the additivity and is naturally possessed by one-dimensional (spatial) systems. The calculation of PEE using *ALC proposal* in higher dimensions is ambiguous. This is because there is no natural ordering of the configuration of the subsets inside a given region. Still, in those cases, the *ALC proposal* only applies for configurations with high symmetry such that the contour function only depends on one specific coordinate,

---

[1]The entanglement contour has been largely explored in free theories [47–54]. In the purview of holography, one can consider the finer description offered by the Ryu-Takanayagi prescription to relate points on the minimal surface to the corresponding boundary points [27,55]. Explicit constructions of the entanglement contour and the PEE in terms of bit threads [56] are provided in [57–60] (also see [27,28,54,55]).

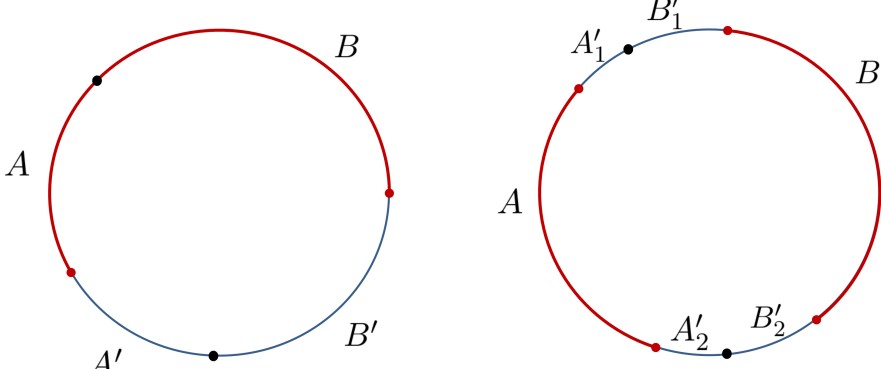

Figure 1: The purification of a region $AB$ is given by $ABB'A'$ for adjacent (left) and non-adjacent (right) intervals. When the intervals are adjacent the auxiliary system $A'B'$ consists of $A'B' = A' \cup B'$. One the other hand, for the non-adjacent case, the auxiliary system $A'B'$ consists of $A'B' = A'_1 \cup B'_1 \cup A'_2 \cup B'_2$.

which can be used as a direction to define a natural ordering. See [62] for more discussion in higher dimensions.

The *entanglement contour* has been studied extensively in the context of the evolution of entanglement [47, 53, 58, 63, 64]. PEE is rather a quasi-local measure of entanglement and gained significant importance in the context of holography [27, 55] as it provides a finer description between quantum entanglement of the boundary theory and the geometry of the bulk [55]. PEE can also be extended to define the dual to the entanglement wedge cross-section. This is achieved by imposing some balance conditions [1], which will be our primary focus on this paper. This balanced PEE can be shown to be applicable for generic purification and naturally incorporates the reflected entropy [22] as a specific case. More details on PEE, especially a first law-like version of the *entanglement contour* and its role in recently proposed island proposal can be found in [63, 65, 66].

## 2.2 The balanced partial entanglement entropy

Based on previous discussions, we introduce the so-called balanced partial entanglement entropy (BPE) [1].

- Consider a bipartite system $\mathcal{H}_A \otimes \mathcal{H}_B$, equipped with the density matrix $\rho_{AB}$. Let us further consider a purified system $ABA'B'$ with a pure state $|\psi\rangle$ which is purified by an auxiliary system $A'B'$, so that $\text{Tr}_{A'B'}|\psi\rangle\langle\psi| = \rho_{AB}$. Then we consider the special partition of the auxiliary system $A'B' = A' \cup B'$ that satisfies the following *balance requirements*

$$\text{balance requirements}: \qquad s_{AA'}(A) = s_{BB'}(B), \qquad s_{AA'}(A') = s_{BB'}(B'). \qquad (9)$$

  With the balance requirements satisfied, the BPE$(A:B)$ is just given by the PEE $s_{AA'}(A)$, i.e.,

$$\text{BPE}(A:B) = s_{AA'}(A)|_{\text{balance}}. \qquad (10)$$

However, in general, the partition that satisfies the balance requirements is not unique. To emphasize this point and to remove the ambiguity, we further impose the condition of *minimality*. This ensures that among all the possible partitions, we need to pick the partitions

such that $s_{AA'}(A)$ is minimized. The partition has been done so that $A'$ is settled to be as far from $B$ as possible while $B'$ is settled to be as far from $A$ as possible. Also, there should be no embedding between $B'$ and $A'$. See Fig.1 for an explanation and refer to [1] for more details.

Since $S_{AA'} = S_{BB'}$, we can consider one of the conditions in (9) as independent. Furthermore, we can write the entanglement entropies as[2]

$$s_{AA'}(A) = \mathcal{I}_{AB} + \mathcal{I}_{AB'}, \qquad s_{BB'}(B) = \mathcal{I}_{BA} + \mathcal{I}_{BA'}. \qquad (11)$$

Since $\mathcal{I}_{AB}$ is independent from the purifications, and $\mathcal{I}_{AB} = \mathcal{I}_{BA}$, the balance conditions (9) can also be rephrased as

$$\text{balance requirement}: \qquad \mathcal{I}_{AB'} = \mathcal{I}_{BA'}. \qquad (12)$$

Later we will also refer to the PEE $\mathcal{I}_{AB'}$ or $\mathcal{I}_{BA'}$ as the *crossing PEE* of $|\psi\rangle$.

In the case where $A$ and $B$ are not adjacent, the complement $A'B'$ becomes disconnected. In such a case, one can separate out the disconnected regions into further subregions as $A' = A_1' \cup A_2'$ and $B' = B_1' \cup B_2'$. As a result, they can be considered in pairs (see Fig.1 (right)), and the balance conditions should be imposed on both of the pairs, leading to the generalized balance conditions for the disconnected regions as [1]

$$s_{AA'}(A) = s_{BB'}(B), \quad s_{AA'}(A_1') = s_{BB'}(B_1'), \quad s_{AA'}(A_2') = s_{BB'}(B_2'). \qquad (13)$$

Similar to the adjacent case, here, two conditions are independent, which is enough to determine the exact partitions.

We conclude this section by stating some inequalities satisfied by the BPE [1]

$$1. \quad \text{BPE}(A:B) \leq \min(S_A, S_B). \qquad (14)$$

$$2. \quad \text{BPE}(A:B) \geq \frac{1}{2} I(A:B). \qquad (15)$$

$$3. \quad \text{BPE}(A:B) + \text{BPE}(A:C) \geq \text{BPE}(A:BC). \qquad (16)$$

$$4. \quad \text{BPE}(A:BC) \geq \text{BPE}(A:B). \qquad (17)$$

The inequality 1 holds for both of the holographic and non-holographic cases. The property 2 holds in holographic theories [1], for both adjacent and non-adjacent cases. The proof relies on the monogamy of the mutual information [67]. For non-holographic theories, the validity of property 2 is not clear in the case of non-adjacent intervals. Inequality 3 follows from inequality 2. Finally, inequality 4 holds for generic theories.

## 3 The BPE for purifications from Euclidean path-integral

In this section, we construct a class of purifications for the mixed state $\rho_{AB}$ using the Euclidean path-integral [68,69]. Then we study the BPE for these purifications and find that the BPE is independent of this class of purifications.

### 3.1 The Euclidean path-integral and its optimization

Here we follow the steps in [68,69] to prepare pure states from the optimization of Euclidean path-integrals. As shown in Fig. 2, we first consider the adjacent-interval case where $AB$ is

---

[2]Form now onwards we use the shorthand notation $\mathcal{I}_{AB}$ for $\mathcal{I}(A, B)$.

connected and embedded in a pure state $|\Psi\rangle$ of a CFT. The mixed state we consider is just the reduced density matrix $\rho_{AB}$ of the region $AB$

$$\rho_{AB} = \text{Tr}_{A'B'}|\Psi\rangle\langle\Psi|. \tag{18}$$

Consider the CFT on a Euclidean flat space $\mathbb{R}^2$

$$ds^2 = dz^2 + dx^2, \tag{19}$$

where $-z \equiv \tau$ is the Euclidean time.

One prepares the ground-state wave functional for CFTs on this metric i.e., on two-dimensional flat space by computing the Euclidean path integral

$$\Psi_{\text{CFT}}(\tilde{\varphi}(x)) = \int \left( \prod_x \prod_{\epsilon < z < \infty} D\varphi(z,x) \right) e^{-S_{\text{CFT}}(\varphi)} \prod_x \delta\left(\varphi(\epsilon,x) - \tilde{\varphi}(x)\right). \tag{20}$$

The path-integral optimization aims to prepare a ground state wavefunctional by integrating over a different geometry such that the ground state prepared over the new geometry is proportional to the ground state prepared over the flat geometry. The procedure works in any general dimension. However, in two dimensions, the procedure enjoys significant simplification as in this case, any metric can be written in a diagonal form via a coordinate transformation

$$ds^2 = e^{2\phi(z,x)}\left(dz^2 + dx^2\right), \qquad e^{2\phi(z=\epsilon,x)} = 1. \tag{21}$$

The second condition is the boundary condition imposed on $z = \epsilon$, where the state is being prepared and where the metric of both geometries coincide.

After the Weyl transformation, the metric is now described by the scalar field $\phi(z,x)$. With the universal ultraviolet (UV) cutoff $\epsilon$, the measure of quantum fields $\varphi$ in the CFT changes under the Weyl transformation [68]

$$[D\varphi]_{g_{ab}=e^{2\phi}\delta_{ab}} = e^{S_L(\phi)-S_L(0)}[D\varphi]_{g_{ab}=\delta_{ab}}, \tag{22}$$

where $S_L(\phi)$ is the Liouville action [69]

$$S_L[\phi] = \frac{c}{4\pi} \int_{-\infty}^{\infty} dx \int_{\epsilon}^{\infty} dz \left((\partial_x \phi)^2 + (\partial_z \phi)^2 + \mu e^{2\phi}\right). \tag{23}$$

Here $\mu$ is the potential and $c$ is the central charge. Furthermore, since the action $S_L(\varphi)$ is invariant under the Weyl transformation and the boundary condition for the scalar field at $z = \epsilon$ agrees with the original one, the ground-state wave function $\Psi_{g_{ab}=e^{2\phi}\delta_{ab}}$ computed from path-integral with the Weyl transformed metric (21) is proportional to the original one with the flat metric. In other words

$$\Psi_{g_{ab}}(\tilde{\varphi}(x)) = e^{S_L(\phi)-S_L(0)}\Psi_{\delta_{ab}}(\tilde{\varphi}(x)), \tag{24}$$

which implies that the state $\Psi_{g_{ab}}$ is still the same CFT vacuum state (up to the proportionality constant).

Thus, our task is to consider the special configuration of $\phi(z,x)$ that minimizes the Liouville action $S_L(\phi)$ using the path-integral optimization. The optimization is equivalent to minimizing the normalization $e^{S_L(\phi)}$ of the wavefunctional. This can be achieved by solving the equations of motion for the Liouville action (23), whose general solution is given by

$$e^{2\phi} = \frac{4A'(w)B'(\bar{w})}{(1-A(w)B(\bar{w}))^2}, \tag{25}$$

where $w = z + ix$ and $\bar{w} = z - ix$. For the boundary condition in (21), the explicit solution is given by $A(w) = w, B(\bar{w}) = -1/\bar{w}$, hence

$$e^{2\phi} = \frac{\epsilon^2}{z^2}, \tag{26}$$

which exactly gives the metric of the Poincaré patch of AdS$_3$. It is worth mentioning that the authors of [68] interpret this as a continuous limit of the conjectured relation between tensor networks and AdS/CFT correspondence. They also suggest that the optimization is analogous to the estimation of the computational complexity (see also [69]). The reader can refer to [68,69] for further details about path-integral optimization.

When setting boundary conditions for $\phi$ on the whole time slice, the optimization will not change the state we compute. However, when we consider the reduced density matrix of a sub-region and whole time slice as a purification for this region, we will only set boundary conditions on the region. In this case, performing the path-integral optimization will give us a specific configuration for the scalar field on the complement at $z = \epsilon$, which corresponds to a special purification for the region [70].

## 3.2 Purifications for adjacent intervals

### 3.2.1 Purifications from path-integrals without optimization

Here we follow the method outlined in [70]. We consider an interval $[a, b]$ on an infinitely long line. The interval is decomposed into two sub-intervals

$$A = [a, p], \quad B = [p, b], \quad \text{where } -\infty < a < p < b < \infty. \tag{27}$$

Here $A$ and $B$ are adjacents. Originally we introduced a uniform cutoff $\epsilon$ on the line and considered a discretization of the Euclidean path-integral for preparing the vacuum state of the system. The mixed state $\rho_{AB}$ is defined from this CFT vacuum by tracing out the complement of $AB$. The point $Q$ with coordinate $q$ divides the complement into two subsystems $A'$ and $B'$.

We cut the interval $AB$ open, then the path-integral representation of the density matrix $\rho_{AB}$ is given by the path-integral on this cut manifold with the imposed boundary condition

$$e^{2\phi(z=\epsilon,x)} = 1, \qquad a \le x \le b \tag{28}$$

on the upper and lower edge of the slit $AB$. Here we want to generate pure states as purifications for $\rho_{AB}$ from path-integral. A class of the purifications can be achieved from path-integral by setting different boundary conditions for $\phi(z, x)$ at $z = \epsilon$ on $A'B'$. These classes of purifications are still the vacuum state of the CFT but settled on different manifolds. The difference is that the lattice cutoff on $A'B'$ is no longer a uniform constant $\epsilon$. Instead it has spatial dependence controlled by the boundary condition for $\phi(z, x)$ on $A'B'$.

It was shown in [70] that the entanglement entropy for an arbitrary interval covering the region $(x_1, x_2)$ in the purifications $\Psi_{\text{CFT}}^{\phi}$ is given by performing a scale transformation of the standard formula for entanglement entropy

$$S_{EE} = \frac{c}{3} \log\left(\frac{x_2 - x_1}{\epsilon}\right) + \frac{c}{6}\phi(x_2) + \frac{c}{6}\phi(x_1). \tag{29}$$

Then the entanglement entropy for the intervals in Fig. 2 are, for example, given by,

$$S_A = \frac{c}{3} \log\left(\frac{p-a}{\epsilon}\right) + \frac{c}{6}\phi(a) + \frac{c}{6}\phi(p), \quad S_{A'} = \frac{c}{3} \log\left(\frac{q-a}{\epsilon}\right) + \frac{c}{6}\phi(a) + \frac{c}{6}\phi(q),$$
$$S_B = \frac{c}{3} \log\left(\frac{b-p}{\epsilon}\right) + \frac{c}{6}\phi(p) + \frac{c}{6}\phi(b), \quad S_{B'} = \frac{c}{3} \log\left(\frac{q-b}{\epsilon}\right) + \frac{c}{6}\phi(b) + \frac{c}{6}\phi(q). \tag{30}$$

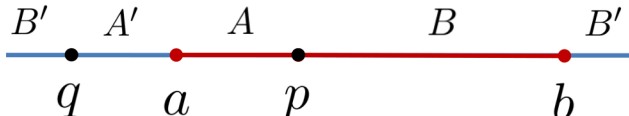

Figure 2: Single interval optimization. (a) The full interval $AB = [a, b]$ is denoted by the red line. The coordinate for the partition point of $AB$ is $p$ and the one for $A'B'$ is $q$.

For the mixed state $\rho_{AB}$ we choose, the scalar fields should vanish inside $AB$ due to the boundary conditions, hence

$$\phi(a) = \phi(b) = \phi(p) = 0. \tag{31}$$

Now we consider an arbitrary purification from the path-integral, which means we will not specify the boundary condition for $\phi$ on $A'B'$ at the beginning. Then we impose the balance condition [1]

$$S_A - S_B = S_{A'} - S_{B'}. \tag{32}$$

Substituting the entanglement entropies (30) into the above balance condition, we get the partition for $A'B'$ with the position of $Q$ settled at

$$q = \frac{2ab - (a+b)p}{a+b-2p}. \tag{33}$$

When the balance condition is satisfied, we substitute the above $q$ in to the PEE $s_{AA'}(A)$ to find the BPE$(A : B)$, which is given by

$$\begin{aligned}\text{BPE}(A : B) = s_{AA'}(A)|_{\text{balance}} &= \frac{1}{2}\left(S_{AA'} + S_A - S_{A'}\right)|_{\text{balance}} \\ &= \frac{c}{6}\log\left(\frac{2(p-a)(b-p)}{\epsilon(b-a)}\right). \end{aligned} \tag{34}$$

Note that the above BPE$(A : B)$ is independent from the scalar field on $A'B'$, which implies that the BPE is independent from the class of purifications determined by the boundary conditions of $\phi(z, x)$ on $A'B'$. For the case where the purification is just the vacuum state dual to the global or Poincaré AdS$_3$ (i.e., $\phi(z = \epsilon, x) = 0$ on the whole time slice), the above BPE is just the area of the EWCS (see Fig.3 and [70] for more details). In all these purifications the BPE captures exactly the same class of correlation between $A$ and $B$ as the EWCS.

Then we calculate the crossing PEE for this class of purifications with the balance condition satisfied. It can be easily computed as

$$\begin{aligned}\mathcal{I}_{AB'}|_{\text{balance}} = s_{AA'B}(A)|_{\text{balance}} &= \frac{1}{2}\left(S_{AA'} + S_{AB} - S_{A'} - S_B\right)|_{\text{balance}} \\ &= \frac{c}{6}\log 2, \end{aligned} \tag{35}$$

which is also independent from the scalar field. Furthermore, it is given by a constant independent from the partition of $AB$ as well as the details of the CFT. This constant was previously found in [1] for the vacuum state duals to pure AdS$_3$.

We stress that in the previous discussion, the partition (33) of $A'B'$ determined by the balance conditions is exactly the partition that minimizes $S_{AA'}$ when path-integral is optimized [70]. This implies that the balanced condition could also be a procedure to minimize certain kinds of correlations, as in the case of the purification from optimized path-integral. We will confirm this expectation in the next section.

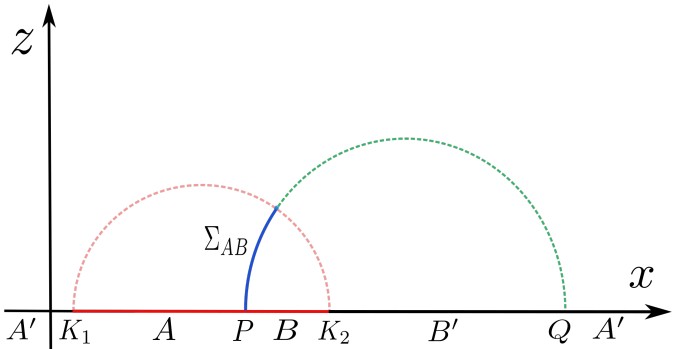

Figure 3: The $x$ coordinates for points $K_1$, $P$, $K_2$ and $Q$ are $a, p, b, q$. The PEE is given by $s_{AA'}(A) = s_{BB'}(B)$ which is dual to the EWCS $\Sigma_{AB}$ (shown by the blue line).

### 3.2.2 Purification from the optimized path-integral

We now consider the special purification determined by path-integral optimization. Following [70], we first perform the following conformal transformation that maps the interval $[a, b]$ to an infinitely long line

$$u = \sqrt{\frac{x-a}{b-x}} \, . \tag{36}$$

Then the boundary condition and the optimization in the $u$ space is exactly the same as the simple case for the vacuum state. The optimized metric is then written as [70]

$$ds^2 = \frac{\epsilon^2}{\tau^2} \, du \, d\bar{u} = \frac{\epsilon^2}{\tau^2} \frac{(b-a)^2}{4|b-y|^3|y-a|} \, dy \, d\bar{y} = e^{2\phi} \, dy \, d\bar{y} \, . \tag{37}$$

Here we have regularized the coordinate $\tau$ as $-\infty < \tau < -\epsilon$ with $\epsilon > 0$. The scalar field $\phi(z, x)$ at the time slice $\tau = -\epsilon$ after the optimization is then given by

$$\phi(x) = \begin{cases} 0 & \text{for} \quad a \leq x \leq b \, , \\ \log\left(\frac{\epsilon(b-a)}{2(x-a)(x-b)}\right) & \text{for} \quad x > b \text{ or } x < a \, . \end{cases} \tag{38}$$

These are obtained from the optimized metric (37). Note the fact that the points $P$ and $Q$ get mapped into the coordinates $u_P = -\sqrt{(p-a)/(b-p)}$ and $u_Q = i\sqrt{(q-a)/(q-b)}$ respectively [70]. We call the pure state with boundary conditions (38) the optimized purification.

It is worth mentioning that path-integral optimization provides a useful tool to evaluate the EoP [70]. In the optimized purification, all the entanglement entropies can be calculated via (29). If we minimize $S_{AA'}$ by choosing a suitable $q$ that partitions $A'B'$, the minimized $S_{AA'}$ coincides with the EWCS of $\rho_{AB}$ (3.2.2). If the correspondence between the EoP and EWCS is valid, then the optimized purification is exactly the one that minimizes $S_{AA'}$ under a suitable partition, and the $E_p(A, B)$ is just the minimized $S_{AA'}$.

Interestingly, the suitable partition that minimizes $S_{AA'}$ is given by (33), which is exactly the solution to the balance condition (32). Furthermore, as we have shown that the BPE $(A : B)$ also yields the area of the EWCS and is independent of the boundary conditions for the scalar field in the optimized purification with $q$ given by (33), we have

$$S_{AA'} = \text{BPE}(A : B) = s_{AA'}(A) = \frac{\text{Area}(\Sigma_{AB})}{4G_N} \, . \tag{39}$$

$$B_2' \quad A_2' \quad A \quad A_1' \ B_1' \quad B \quad B_2'$$
$$q_2 \quad a \quad b \ q_1 \ c \quad d$$

Figure 4: Double interval optimization.

This implies that the contribution to $S_{AA'}$ from $A'$ is zero, *i.e.*,

$$s_{AA'}(A') = \mathcal{I}_{A'B} + \mathcal{I}_{A'B'} = 0. \tag{40}$$

Since the balanced condition is satisfied, according to (35) we have $\mathcal{I}_{A'B} = c/6 \log 2$, which implies

$$\mathcal{I}_{A'B'} = \frac{1}{2}I(A':B') = -\frac{c}{6}\log 2. \tag{41}$$

One can further check the above result with a direct calculation in the optimized purification

$$
\begin{aligned}
\mathcal{I}_{A'B'} = s_{A'AB}(A') &= \frac{1}{2}(S_{A'AB} + S_{A'} - S_{AB}) \\
&= \frac{c}{6}\left(\log\left[\frac{(q-a)(q-b)}{(b-a)\epsilon}\right] + \phi(q)\right) \\
&= -\frac{c}{6}\log 2,
\end{aligned}
\tag{42}
$$

where in the last line we substituted in the partition (33) and the scalar field after optimization (38). This is puzzling because the negative PEE and mutual information in the optimized purification breaks the Araki-Lieb inequality.

### 3.3 Purifications for non-adjacent intervals

Next, we consider the case when $A$ and $B$ are not adjacent. More explicitly, we will consider the case shown in Fig. 4, where

$$A = [a, b], \quad B = [c, b], \quad \text{where} \quad -\infty < a < b < c < d < \infty. \tag{43}$$

Also in this case the complement region $A'B'$ consists of two disconnected regions. We use $q_1$ and $q_2$ to partition them into $A_1' \cup B_1'$ and $A_2' \cup B_2'$.

As in the adjacent case, the path-integral under different boundary conditions for $\phi(z = \epsilon, x)$ on $A'B'$ define a class of purifications, which are the vacuum state of the same CFT but on different manifolds. For any configurations of $\phi$ on $A'B'$, we can calculate the entanglement entropies for intervals via (29). Using the ALC proposal to calculate the PEEs, the balance conditions (13) equate the following to equations [1]

$$S_{A_1'} - S_{B_1'} = S_{AA_2'} - S_{BB_2'}, \quad S_{A_2'} - S_{B_2'} = S_{AA_1'} - S_{BB_1'}. \tag{44}$$

Using (29) to calculate the entanglement entropies, we find that all the scalar fields cancel with each other, and the balance conditions are solved by

$$
\begin{aligned}
q_1 &= \frac{\sqrt{(a-b)(a-c)(b-d)(c-d)} + ad - bc}{a-b-c+d}, \\
q_2 &= -\frac{\sqrt{(a-b)(a-c)(b-d)(c-d)} - ad + bc}{a-b-c+d}.
\end{aligned}
\tag{45}
$$

We then calculate the BPE under the above balance condition, which is again independent from the scalar fields and given by

$$
\begin{aligned}
\mathrm{BPE}(A:B) &= s_{AA'}(A)|_{\mathrm{balance}} \\
&= \frac{1}{2}(S_{AA'_1} + S_{AA'_2} - S_{A'_1} - S_{A'_2})|_{\mathrm{balance}} \\
&= \frac{c}{6}\log\left(\frac{a(b+c-2d)+2\sqrt{(a-b)(a-c)(d-b)(d-c)}+d(b+c)-2bc}{(a-d)(b-c)}\right).
\end{aligned}
$$
(46)

Though the above result looks a bit complicated, in the context of AdS/CFT, it gives the EWCS when the entanglement wedge of $AB$ is connected.

Let us consider the following symmetric case where

$$
A = [1, 1/x], \quad B = [-1/x, -1], \quad \text{where } 0 < x < 1.
$$
(47)

Substituting the above parameters into the BPE (46), we find that the partition respects the reflection symmetry and is given by $q_1 = 0$, $q_2 \to \infty$, and the BPE is given by the following simple formula

$$
\mathrm{BPE}(A:B) = -\frac{c}{6}\log x,
$$
(48)

which exactly matches with the leading term of EoP obtained through the minimization of $S_{AA'}$ in the optimized purification in [70]. It is worth noting, however, that the calculation in [70] is only valid when $x$ is small.

## 4 Balance conditions as extremal conditions

In this section, we demonstrate that the BPE could be a proper measure of the total intrinsic correlation between $A$ and $B$ in the mixed state $\rho_{AB}$. In the following, we characterize all the correlations in terms of the PEE and denote this total intrinsic correlation between $A$ and $B$ as $\mathcal{C}(A,B)$. Let us forget for a moment about the BPE and start from the unknown measure $\mathcal{C}(A,B)$ with the following expectations:

- $\mathcal{C}(A,B)$ is intrinsic to $A$ and $B$, and hence should be independent of the purifications.

- $\mathcal{C}(A,B)$ should give the reflected entropy when we consider the canonical purification.

- For a purification with holographic dual, $\mathcal{C}(A,B)$ should be given by the EWCS.

Later we will show how to determine the expression of $\mathcal{C}(A,B)$ in terms of PEE.

### 4.1 Minimizing the crossing correlation for adjacent cases

Let us first consider the simple example where $A$ and $B$ are adjacents. When a purification is given, it is not clear why the balance condition for the PEE $s_{AA'}(A) = s_{BB'}(B)$ can help us identify the total intrinsic correlation $\mathcal{C}(A,B)$. We perform a decomposition of the complement $A'B' = A' \cup B'$ so that the purification consists of four different regions whose correlation structure can be characterized by the following six different PEEs:

$$
\mathcal{I}_{AB}, \qquad \mathcal{I}_{AB'}, \qquad \mathcal{I}_{AA'}, \qquad \mathcal{I}_{BB'}, \qquad \mathcal{I}_{A'B}, \qquad \mathcal{I}_{A'B'}.
$$
(49)

Some of the PEE and entanglement entropies are determined by the density matrix $\rho_{AB}$, hence are independent of the purifications. These include the entanglement entropies $S_{AB}$, $S_A$ and $S_B$, which can be written as a collection of PEEs

$$
\begin{aligned}
S_A &= \mathcal{I}_{AB} + \mathcal{I}_{AB'} + \mathcal{I}_{AA'}, \\
S_B &= \mathcal{I}_{AB} + \mathcal{I}_{BB'} + \mathcal{I}_{BA'}, \\
S_{AB} &= \mathcal{I}_{AB'} + \mathcal{I}_{AA'} + \mathcal{I}_{BA'} + \mathcal{I}_{BB'}.
\end{aligned}
\tag{50}
$$

The above identities can be derived using the ALC proposal of PEE in (8). The main implication is that the PEE sums up to give the exact entropies in the left-hand sides (LHSs). For example, the first one computes the contribution from $B$, $B'$, and $A'$ to the region $A$.

The independence from purifications for $\mathcal{I}_{AB}$ can be derived from the purification independence of the above entanglement entropies. Also, we find the purification independence of the following two linear combinations of the PEE, which we denote as $\mathcal{P}_1$ and $\mathcal{P}_2$

$$
1)\ \mathcal{I}_{AB'} + \mathcal{I}_{AA'} = \mathcal{P}_1, \qquad 2)\ \mathcal{I}_{BB'} + \mathcal{I}_{BA'} = \mathcal{P}_2.
\tag{51}
$$

In the following, we analyze which part of the six PEEs may contribute or relate to the intrinsic correlation $\mathcal{C}(A, B)$:

- The PEE $\mathcal{I}_{AB}$ is a direct and intrinsic measure of certain correlations between $A$ and $B$, hence should be included in $\mathcal{C}(A, B)$. Furthermore, in the special case where $A$ and $B$ are adjacent, $\mathcal{I}_{AB} = I(A : B)/2$.

- The crossing PEEs $\mathcal{I}_{AB'}$ and $\mathcal{I}_{BA'}$ that cross the four regions contribute partially to $\mathcal{C}(A, B)$. They also contribute to the correlation between $AB$ and $A'B'$.

- The PEEs $\mathcal{I}_{AA'}$ and $\mathcal{I}_{BB'}$ mainly sustain the entanglement between $AB$ and $A'B'$. They may also partially contribute to $\mathcal{C}(A, B)$.

- The PEE $\mathcal{I}_{A'B'}$ definitely has no contribution to $\mathcal{C}(A, B)$. It is possible to eliminate this correlation part via unitary transformations inside $A'B'$.

How can we extract the correlation $\mathcal{C}(A, B)$ from the PEEs with a given purification? Unlike the EoP, we are not going to minimize over all the possible purifications to find the minimal $S_{AA'}$. Also, unlike the reflected entropy $S_R(A, B)$, we will not restrict ourselves to the canonical purification, which needs the explicit density matrix of $\rho_{AB}$. When the purification is fixed, the only parameter we can adjust is the partition of the complement region $A'B'$, i.e., the position of the partition point $Q$ in this case and all the PEEs except $\mathcal{I}_{AB}$ will be affected by the position of $Q$. Suppose that we start at some point near $A$, then move $Q$ towards $B$ by a small distance $dq$. This operation expands $A'$ and shrinks $B'$, hence increases $\mathcal{I}_{BA'}$ and decreases $\mathcal{I}_{AB'}$. At the same time, we keep in mind that the combinations (51) do not depend on $Q$. Due to the additivity of the PEE, the changing of the PEEs can be explicitly described in the following way:

$$
\begin{aligned}
\mathcal{I}_{BA'} &\to \mathcal{I}_{BA'} + \mathcal{I}_{B(dq)}, & \mathcal{I}_{BB'} &\to \mathcal{P}_2 - \mathcal{I}_{BA'} - \mathcal{I}_{B(dq)}, \\
\mathcal{I}_{AB'} &\to \mathcal{I}_{AB'} - \mathcal{I}_{A(dq)}, & \mathcal{I}_{AA'} &\to \mathcal{P}_1 - \mathcal{I}_{BA'} + \mathcal{I}_{A(dq)}.
\end{aligned}
\tag{52}
$$

We do not need to discuss the change of $\mathcal{I}_{A'B'}$ since it will be excluded from $\mathcal{C}(A, B)$.

When we say $Q$ is close to $A$, we mean $\mathcal{I}_{A(dq)} > \mathcal{I}_{B(dq)}$. If we move $Q$ towards $B$, we will first arrive at a balance point where

$$
\mathcal{I}_{A(dq)} = \mathcal{I}_{B(dq)},
\tag{53}
$$

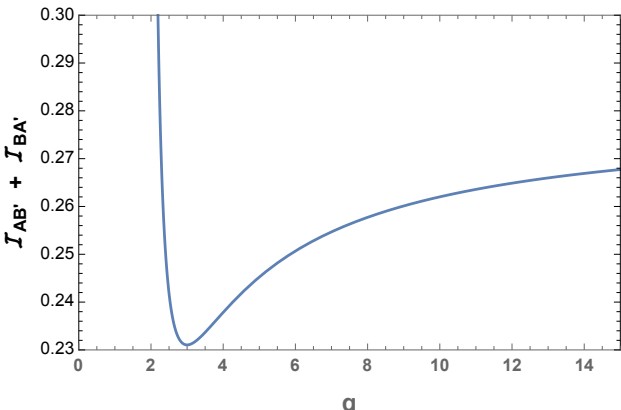

Figure 5: Here we set $c = 1$, $a = 0$, $b = 2$, and $p = 3/2$. The curve shows how the sum of the crossing PEEs $\mathcal{I}_{AB'} + \mathcal{I}_{BA'}$ varies with respect to $q$. The sum reaches its minimal value of $(\log 2)/3 = 0.231049$ at $q = 3$, which is the balance point.

then we enter the region close to $B$ where $\mathcal{I}_{A(\mathrm{d}q)} < \mathcal{I}_{B(\mathrm{d}q)}$. Then it is natural to consider the combination $\mathcal{I}_{AB'} + \mathcal{I}_{BA'}$ which decreases at first, reaches its minimal value at the balance point, then increase as $Q$ moves further towards $B$. One can check this with an explicit calculation of this combination

$$
\begin{aligned}
\mathcal{I}_{AB'} + \mathcal{I}_{BA'} &= \frac{1}{2}\left(S_{AA'} + S_{AB} - S_{A'} - S_B\right) + \frac{1}{2}\left(S_{AB} + S_{BB'} - S_A - S_{B'}\right) \\
&= \frac{c}{6}\log\left(\frac{(a-b)^2(p-q)^2}{(a-p)(a-q)(p-b)(b-q)}\right),
\end{aligned}
\tag{54}
$$

where the PEEs are calculated by the ALC proposal (8). The above expression reaches its minimal value $c/3 \log 2$ at the saddle point (33), where the balance condition is satisfied (see Fig.5). Since the minimization and the balance condition coincide with each other, we conclude

$$
(\mathcal{I}_{AB'} + \mathcal{I}_{BA'})|_{\text{minimized}} = 2\mathcal{I}_{AB'}|_{\text{balance}} = 2\mathcal{I}_{A'B}|_{\text{balance}} = \frac{c}{3}\log 2.
\tag{55}
$$

The above analysis shows that if we exclude the contribution from $\mathcal{I}_{A'B'}$, then the sum of the crossing correlations between $AB'$ and $BA'$ is minimized at the balance point. This looks similar to the definition of EoP. However, for an arbitrary purification, $S_{AA'} - \mathcal{I}_{A'B'}$ still captures more correlations than $\mathcal{C}(A, B)$. If the crossing PEEs contribute to $\mathcal{C}(A, B)$, they also contribute to the correlation $\mathcal{C}(A', B')$ in a similar sense. This implies that the crossing PEE can only contribute partially to $\mathcal{C}(A, B)$. Before we determine the $\mathcal{C}(A, B)$ in terms of the PEE, we still need to answer the following two questions:

- How are the minimized crossing PEEs assigned to the correlations $\mathcal{C}(A, B)$ and $\mathcal{C}(A', B')$ respectively?

- When the crossing PEEs are minimized, do the PEEs $\mathcal{I}_{AA'}$ and $\mathcal{I}_{BB'}$ contribute to $\mathcal{C}(A, B)$?

## 4.2 The universality of the crossing PEE and the Markov gap

The answers for the above two questions can be found from our expectation that $\mathcal{C}(A, B)$ should give the reflected entropy for the canonical purification and be independent from any particular purification. This means that the different value between $\mathcal{C}(A, B)$ and the intrinsic $\mathcal{I}_{AB}$ should be purification-independent and given by

$$
\mathcal{C}(A, B) - \mathcal{I}_{AB} = \frac{1}{2}S_R(A : B) - \mathcal{I}_{AB}.
\tag{56}
$$

In the adjacent case where $\mathcal{I}_{AB} = I(A : B)/2$, the above difference is just given by the so called Markov gap [71–73], which is defined as the difference between the half of the reflected entropy and the mutual information[3]

$$h(A, B) = \frac{1}{2} \left( S_R(A : B) - I(A : B) \right). \tag{57}$$

Remarkably, explicit calculations in 2d CFTs show that the Markov gap is given by a universal constant $c/6 \log 2$ for the adjacent case. However, the equation (56) suggests a generalization of the Markov gap from the canonical purification to more generic purifications, where it is generalized to be the difference between the correlation $\mathcal{C}(A, B)$ and $\mathcal{I}_{AB}$.

We argue that the proper interpretation for $\mathcal{C}(A, B) - \mathcal{I}_{AB}$ is just the minimized (or balanced) crossing PEE

$$\mathcal{C}(A, B) - \mathcal{I}_{AB} = \frac{1}{2} \left( \mathcal{I}_{AB'} + \mathcal{I}_{A'B} \right)|_{\text{minimized}} = \mathcal{I}_{AB'}|_{\text{balance}} = \mathcal{I}_{A'B}|_{\text{balance}}, \tag{58}$$

which directly suggest that the correlation $\mathcal{C}(A, B)$ is exactly given by the $\text{BPE}(A : B)$

$$\mathcal{C}(A, B) = \mathcal{I}_{AB} + \mathcal{I}_{AB'}|_{\text{balance}} = \text{BPE}(A : B). \tag{59}$$

The reason we propose (58) is the following: firstly, it was shown in [1] that for the canonical purification, the Markov gap coincides with the balanced crossing PEE. In this case, the partition of $A'B'$ automatically satisfies the balance condition and the reflected entropy can be naturally interpreted as the PEE $s_{AA'}(A) = S_{AA'}/2$ due to the symmetry between $A'$ and $A$. Also the reflection symmetry between $AB$ and $A'B'$ implies that the minimized crossing PEEs contribute equally to $\mathcal{C}(A, B)$ and $\mathcal{C}(A', B')$. Hence, at the balance point, only half of the crossing PEEs contribute to $\mathcal{C}(A, B)$. Secondly, for all the purifications we explored, the balanced crossing PEEs are all given by the same constant $c/6 \log 2$, which is the same as the Markov gap. These purifications include:

- The vacuum state of the holographic $\text{CFT}_2$ [1].

- The canonical purification in holographic [1, 71] and several generic 2d CFTs [71] including the Ising CFT, the tricritical Ising CFT, the compactified free boson CFT with different compactification radius. The universality of this constant even extends to $(2 + 1)$-dimensional topological phases [72, 74].

- The class of purifications we obtained from path-integral optimization with different boundary conditions for the metric on the compliment (see section 3).

Like the Markov gap, the balanced crossing PEEs in all the above purifications are independent of both the partition of $AB$ and the length of $AB$. It is then natural to propose that the generalization of the Markov gap for general purifications is just the balanced crossing PEE, which is a universal constant that only depends on the central charge of the CFT. It is independent of not only the purifications but also the details of the mixed state. One can give a naive argument for this universality when the pure state is defined on a circle. In $\text{CFT}_2$, the entanglement entropy of a single interval is given by a universal formula. Since the crossing PEE can be written as a special linear combination of the entanglement entropy for single intervals and all the lengths, cancel with each other when the balance condition is satisfied, and the crossing PEE for CFTs should be given by the same constant. At the balance point, since the PEE $\mathcal{I}_{AB}$ plus the balanced crossing PEE exactly give the reflected entropy and the EWCS, $\mathcal{C}(A, B)$ does not receive any contribution from $\mathcal{I}_{AA'}$ and $\mathcal{I}_{BB'}$.

---

[3]Our notation differs from [71, 72] by a factor of $1/2$.

### 4.3 Minimizing the crossing correlation for non-adjacent cases

Now we consider two non-adjacent intervals in the vacuum of the $CFT_2$ which correspond to the global $AdS_3$. We set the length of the boundary to be $2\pi$, thus the entanglement entropy for a single interval with length $\ell$ is given by $S = \frac{c}{3}\log\left(\frac{2}{\varepsilon}\sin\frac{\ell}{2}\right)$. In this case, the setup for non-adjacent system $AB$ is shown in Fig. 1 (right). The intervals have the following lengths:

$$l_A = 2a, \quad l_B = 2b, \quad l_{A'_1} = 2a_1, \quad l_{B'_1} = 2b_1 \quad l_{A'_2} = 2a_2, \quad l_{B'_2} = 2b_2. \tag{60}$$

We also define the length of the boundary circle to be $2\pi$ and $a_1 + b_1 = \alpha$, thus we have $a_2 + b_2 = 2\pi - 2a - 2b - 2\alpha$. As soon as the length and coordinates of $A$ and $B$ are given, the parameter $\alpha$ is also determined, thus only two parameters remain undetermined. We take them as $a_1$ and $a_2$. Hence, we have

$$b_1 = \alpha - a_1, \qquad b_2 = \pi - \alpha - a - b - a_2. \tag{61}$$

The balance requirements (13) are equivalent to the following equations [1]

$$S_{A'_1} - S_{B'_1} = S_{AA'_2} - S_{BB'_2}, \qquad S_{A'_2} - S_{B'_2} = S_{AA'_1} - S_{BB'_1}, \tag{62}$$

which gives the following equations

$$\frac{\sin[a_1]}{\sin[\alpha - a_1]} = \frac{\sin[a + a_2]}{\sin[\alpha + a + a_2]}, \quad \frac{\sin[a_2]}{\sin[\alpha + a + b + a_2]} = \frac{\sin[a + a_1]}{\sin[a + a_2]}, \tag{63}$$

and determine the position of the partition points $P_1$ and $P_2$. The solution is given by [1]

$$\begin{aligned} a_1 &= \cos^{-1}\left[\frac{\sin(a - \alpha + a_2) + 3\sin(a + \alpha + a_2)}{\sqrt{2}\sqrt{-2\cos(2(a + \alpha + a_2)) - 2\cos(2(a + a_2)) + \cos(2\alpha) + 3}}\right], \\ a + 2a_2 &= \tan^{-1}[\sin(a - b)(\sin(a)\cos(\eta) - \sin(b)) - 2\xi\sin(a)\sin(\eta) \\ &\quad - \sin(a)\sin(\eta)\sin(a - b) - 2\xi\sin(a)\cos(\eta) + 2\xi\sin(b)], \end{aligned} \tag{64}$$

where

$$\xi = \sqrt{\sin(a)\sin(b)\sin(a + \alpha)\sin(\alpha + b)}, \tag{65}$$

$$\eta = a + b + 2\alpha. \tag{66}$$

Now we define three combinations of the crossing PEEs:

$$\begin{aligned} C_1 &\equiv \mathcal{I}_{AB'_1} + \mathcal{I}_{BA'_1} + \mathcal{I}_{AB'_2} + \mathcal{I}_{BA'_2}, \\ C_2 &\equiv \mathcal{I}_{A'_1 B'_2} + \mathcal{I}_{B'_1 A'_2}, \\ C_3 &\equiv C_1 + C_2. \end{aligned} \tag{67}$$

All the above PEEs can be easily calculated by the ALC proposal. The direct generalization of the crossing PEE is $C_1$. It was also verified in [1] that the BPE$(A : B)$ has the following decomposition as in (58)

$$\text{BPE}(A : B) = s_{AA'}(A)|_{\text{balance}} = \mathcal{I}_{AB} + \frac{1}{2}C_1|_{\text{balance}}, \tag{68}$$

where $C_1$ plays exactly the role of the balanced crossing PEE. The above BPE furthermore gives the area of the EWCS [1]. The combination $C_2$ is new compared with the adjacent case, which is correlation crossing the subregions but inside $A'B'$.



Figure 6: Left: $C_1 = C_1(a_1, a_2)$, middle: $C_2 = C_2(a_1, a_2)$, right: $C_3 = C_3(a_1, a_2)$ in the case of $a = \pi/2, b = \pi/4$ and $\alpha = \pi/8$. We find that $C_1, C_3$ has a minimum point and $C_2$ has a saddle point. Note that we have introduced a cutoff as $a_1, a_2$ starts from 0.1 rather than 0.

Then we take the first derivatives of $C_i$ with respect to $a_1, a_2$ and solve the equations

$$\partial_{a_i} C_j = 0, \qquad i = 1, 2; \qquad j = 1, 2, 3. \tag{69}$$

Generally, in the following regions

$$\begin{aligned} 0 &< a_1 < \alpha, \\ 0 &< a_2 < \pi - a - b - \alpha. \end{aligned} \tag{70}$$

$C_1 = C_1(a_1, a_2)$ and $C_3 = C_3(a_1, a_2)$ have a minimum value while $C_2 = C_2(a_1, a_2)$ has a saddle point. See Fig. 6 for a typical example. One can further check that the solutions (64) to the balance conditions also solve the saddle conditions (69) for all the three combinations $C_i$. So, as in the adjacent case, the crossing PEEs are minimized at the balance point.

However, we note that the balanced crossing PEEs, in this case, are no longer a universal constant but depend on the interval sizes of $A$ and $B$. This can be traced back to the fact that the correlation $\mathcal{I}_{AB}$ does not reduce to $I(A : B)/2$ for the non-adjacent intervals. Hence, one needs to be more careful about defining a generalized version of the Markov gap, which is expected to be universal. However, like the Markov gap [73], the above crossing PEES might satisfy certain bounds. When the balanced conditions are satisfied, one can verify that

$$\mathcal{I}_{A_2'(BB_1')} = \mathcal{I}_{B_2'(AA_1')} = \mathcal{I}_{A_1'(BB_2')} = \mathcal{I}_{B_1'(AA_2')} = \frac{c}{6} \log 2, \tag{71}$$

which is because the adjacent configuration is recovered when we consider, for example $\rho_{(AA_2')(BB_2')}$ as a new bipartite system with the two subsystems being $AA_2'$ and $BB_2'$. We will make some brief comments about this in later sections.

## 5 BPE in flat holography and Markov recovery

### 5.1 Holographic entanglement and EWCS in 3d flat holography

Our claim that the BPE captures the total intrinsic correlations in the mixed state should apply to generic theories. In holographic theories beyond AdS/CFT, the BPE should also correspond to the EWCS. Here we conduct an explicit test for our claim in 3d flat holography. In this case, the asymptotic symmetry group is the three-dimensional Bondi-Metzner-Sachs (BMS$_3$) group, which is infinite-dimensional. The correspondence between 3d asymptotic flat space-time and the field theory invariant under the BMS$_3$ group (BMSFT)[4] settled on the null infinity was proposed in [76, 77]. Cardy-like formulas [78–80] are proposed to reproduce the

---

[4]Since the algebra of BMS$_3$ group and the Galilean conformal algebra (GCA) [75] are isomorphic, the duality is also denoted as the GCFT/flat-space correspondence.

Bekenstein-Hawking entropy for the cosmological horizons. More importantly, the geometry for the holographic entanglement entropy is constructed in [80] (see also [55,81,82]), which furthermore inspires the construction of the EWCS in flat spacetime [83]. See also [84,85] for other discussions of entanglement in BMSFTs and [81,86] for further development in 3d flat holography.

For an asymptotically flat 3d spacetime, we characterize the null infinity where the dual BMSFT lives with the coordinates $(u, \phi)$, where the $u$ direction is null, and the $\phi$ direction is spacelike. The generators of the asymptotic symmetries, which forms the $\text{BMS}_3$ group, are the following

$$L_n = u^{n+1}\partial_u + (n+1)u^n\phi\,\partial_\phi\,, \quad M_n = u^{n+1}\partial_\phi. \qquad n = 0, \pm 1, \pm 2 \cdots . \tag{72}$$

The conserved charges satisfy a centrally extended version of the algebra given by

$$
\begin{aligned}
[L_m, L_n] &= (m-n)L_{m+n} + \frac{c_L}{12}(m^3 - m)\delta_{m+n,0}\,, \\
[L_m, M_n] &= (m-n)M_{m+n} + \frac{c_M}{12}(m^3 - m)\delta_{m+n,0}\,, \\
[M_m, M_n] &= 0\,.
\end{aligned}
\tag{73}
$$

Note the two types of central charges $c_L$ and $c_M$ depend on the specific gravity dual. Here we only focus on the BMSFT that duals to the Einstein gravity, which corresponds to the following choice for the central charges [87]

$$c_L = 0\,, \qquad c_M = \frac{3}{G_N}\,. \tag{74}$$

Here we focus on some classical background among the general classical solutions of Einstein gravity without cosmological constant, which take the following form in Bondi gauge [88]

$$ds^2 = 8G_N M\,du^2 - 2\,du\,dr + 8G_N J\,du\,d\phi + r^2 d\phi^2\,. \tag{75}$$

We only discuss the case of the null-orbifold with the parameters $M = J = 0$. This is dual to the zero temperature BMSFT on a plane (an analog of the zero temperature BTZ black hole). One can take a boundary interval $A : \{(u_1, \phi_1), (u_2, \phi_2)\}$ and proceed to calculate the entanglement entropy in this theory. The entanglement entropy of such an interval takes the following simple formula [80,84,85]

$$S_A = \frac{c_L}{6}\ln\left(\frac{u_{12}}{\varepsilon}\right) + \frac{c_M}{6}\left(\frac{\phi_{12}}{u_{12}}\right), \tag{76}$$

where $\phi_{12} = \phi_2 - \phi_1$ and $u_{12} = u_2 - u_1$ and $\varepsilon$ is the (lattice) cutoff. Note that the first term is logarithmic with central charge $c_L$, whereas the second term does not involve any logarithm. Equipped with this result, we will calculate the BPE in BMSFT on some purification.

Another essential ingredient for our discussion is the geometric picture for holographic entanglement entropy [80], which can be used to construct the analogue of the EWCS in 3d flat space. The construction of the EWCS was explicitly studied in [83], with the motivation to establish the duality between the EWCS and the entanglement negativity[5] [91] in 3d flat holography.

The geometric picture for entanglement entropy in flat holography [80] not only contains a bulk spacelike geodesic, but also contains additional null geodesics, which are novel compared with the RT formula in AdS/CFT. This novel geometric picture with null geodesics for

---

[5]See also [89,90] for relevant discussions.

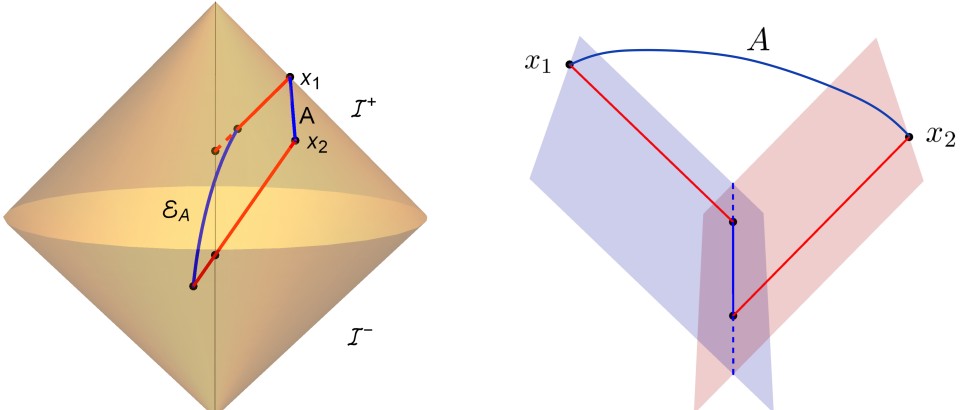

Figure 7: In the left figure, we consider an interval $A$ on the future null infinity $\mathcal{I}^+$. The red lines are the null geodesics $\gamma_{1,2}$ along the $r$ direction emanating from the endpoints $x_{1,2}$. The solid blue line $\mathcal{E}_A$ is the RT surface which is the saddle geodesic with minimal length among all the bulk geodesics connecting $\gamma_{1,2}$. The right figure shows the same geometric picture in Cartesian coordinate coordinates where all the geodesics are straight lines. The two planes are the two normal null hypersurfaces $\mathcal{N}_\pm$ emanating from $\mathcal{E}_A$.

holographic entanglement is argued [55] to be valid for spacetimes with non-Lorentz invariant duals based on a modified version of the Lewkowycz-Maldacena prescription [92]. Let us consider a boundary interval $A$ on the null infinity with endpoints $x_{1,2}$. To each endpoint $x_i$, we associate a null geodesic $\gamma_i$ emanating from it and extending into the bulk. The null geodesic is determined by the requirement that it should follow the bulk modular flow. In the Bondi gauge, they are just null lines along the $r$ direction. The spacelike geodesic $\mathcal{E}_A$ whose length gives the holographic entanglement entropy is just the geodesic that gives the minimal length among all the geodesics whose endpoints are respectively anchored on $\gamma_1$ and $\gamma_2$. For simplicity, we also denote $\mathcal{E}_A$ as the RT surface. See Fig. 7 for the geometric picture in both the Bondi gauge and the Cartesian coordinates. The boundary together with the two normal null hypersurfaces $\mathcal{N}_\pm$ emanating from $\mathcal{E}_A$ enclose a bulk region, which is the analog of the entanglement wedge in flat space.

The EWCS in 3d flat space [83] is defined in a similar way as the EWCS on a time slice of the AdS background. Let us consider the configurations shown in Fig. 8. For the adjacent case where $x_2$ is the endpoint shared by $A$ and $B$, the EWCS is the saddle geodesic whose endpoints $y_b$, and $y_b'$ are anchored on $\mathcal{E}_{AB}$ and $\gamma_2$ respectively. For the non-adjacent case, the entanglement wedge of $AB$ also undergoes a phase transition from the disconnected phase to the connected phase when $A$ and $B$ are close enough. In the connected phase, the EWCS is then given by the saddle geodesic whose endpoints are anchored on $\mathcal{E}_{23}$ and $\mathcal{E}_{14}$. Here, for example, $\mathcal{E}_{23}$ is the RT surface of the interval with endpoints being $x_2$ and $x_3$. The EWCSs are drawn by the solid green lines in Fig. 8, and their lengths are explicitly calculated in [83]. In the following, we will reproduce the EWCS via the BPE in BMSFT.

## 5.2 BPE for adjacent cases and the Markov recovery

First, we consider two adjacent intervals $A : \{(u_1, \phi_1), (u_2, \phi_2)\}$ and $B : \{(u_2, \phi_2), (u_3, \phi_3)\}$. The system is purified with an auxiliary system $A'B'$ partitioned by the point $Q : (u_q, \phi_q)\}$. The

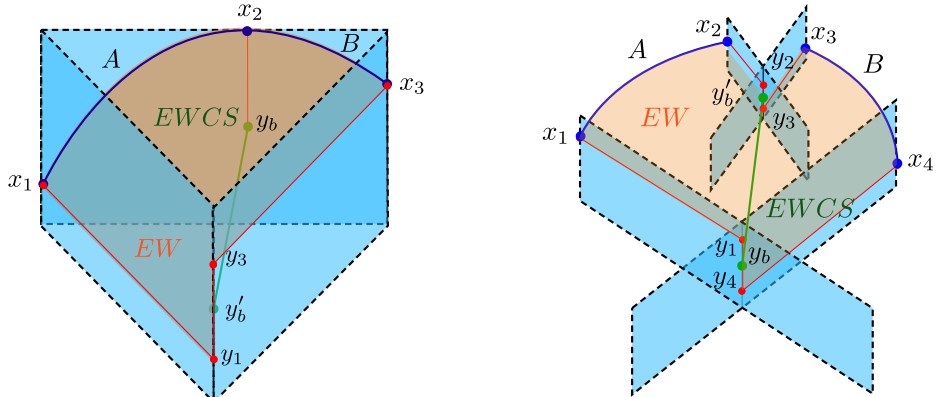

Figure 8: Boundary intervals (left: adjacent, right: non-adjacent) and their entanglement wedge. The red lines emanating from the boundary points $x_i$ are the null lines $\gamma_i$. Here $y_i (= 1, 2, 3, 4)$ are the endpoints of relevant RT surfaces connecting $\gamma_i$. $y_b$ and $y_b'$ are the endpoints of the EWCS (green), which is the saddle geodesic connecting those RT surfaces. In the left figure, $y_b$ is just a point on $\gamma_2$. The figures are inspired by [83].

entanglement entropy for each interval is given by

$$S_A = \frac{c_L}{6} \ln\left(\frac{u_{12}}{\varepsilon}\right) + \frac{c_M}{6}\left(\frac{\phi_{12}}{u_{12}}\right), \quad S_B = \frac{c_L}{6} \ln\left(\frac{u_{23}}{\varepsilon}\right) + \frac{c_M}{6}\left(\frac{\phi_{23}}{u_{23}}\right),$$

$$S_{A'} = \frac{c_L}{6} \ln\left(\frac{u_{q1}}{\varepsilon}\right) + \frac{c_M}{6}\left(\frac{\phi_{q1}}{u_{q1}}\right), \quad S_{B'} = \frac{c_L}{6} \ln\left(\frac{u_{3q}}{\varepsilon}\right) + \frac{c_M}{6}\left(\frac{\phi_{3q}}{u_{3q}}\right), \tag{77}$$

where $\phi_{q1} = \phi_q - \phi_1$ and $u_{q1} = u_q - u_1$ and similarly for $\phi_{3q}$ and $u_{3q}$. As we are considering BMSFT dual to the Einstein gravity, we set $c_L = 0$. The balance condition (32) implies

$$\frac{\phi_{12}}{u_{12}} - \frac{\phi_{23}}{u_{23}} = \frac{\phi_{q1}}{u_{q1}} - \frac{\phi_{3q}}{u_{3q}}. \tag{78}$$

Previously for the $\text{CFT}_2$ case, we defined the pure state on a time slice, and hence only one parameter is needed to be determined. In BMSFT, the entanglement entropy for an interval diverges when the end points are settled on a $u$ slice. Here we consider the covariant configuration where the partition point is characterized by two parameters. However, note that the balance condition is only one equation. Imposing the balance condition, we obtain a line for the partition point $Q : (u_q, \phi_q)$ as

$$\phi_q = \left(\frac{1}{u_{q1}} + \frac{1}{u_{3q}}\right)^{-1}\left(\frac{\phi_{12}}{u_{12}} + \frac{\phi_1}{u_{q1}} - \frac{\phi_{23}}{u_{23}} + \frac{\phi_3}{u_{3q}}\right). \tag{79}$$

One can verify that all the points on the above line give the same BPE. We first calculate the PEE $s_{AA'}(A)$ with $Q$ undetermined

$$s_{AA'}(A) = \frac{1}{2}(S_A + S_{AA'} - S_{A'}) = \frac{c_M}{12}\left(\frac{\phi_{12}}{u_{12}} + \frac{\phi_{q2}}{u_{q2}} - \frac{\phi_{q1}}{u_{q1}}\right). \tag{80}$$

Then we substitute the solution (79) of the balance condition into (80) to obtain the BPE

$$\text{BPE}(A : B) = \frac{c_M}{12}\left(\frac{\phi_{12}}{u_{12}} + \frac{\phi_{23}}{u_{23}} - \frac{\phi_{13}}{u_{13}}\right), \tag{81}$$

which is independent of $u_q$. Upon the substitution $c_M = 3/G_N$, this result exactly matches the EWCS obtained in [83] for adjacent intervals.

Furthermore, we can calculate the crossing PEE at the balance point, which is given by

$$\mathcal{I}_{AB'} = \frac{1}{2}(S_{A'A} + S_{AB} - S_{A'} - S_B) = \frac{c_M}{12} \log\left[\frac{\phi_{q2}}{u_{q2}} + \frac{\phi_{13}}{u_{13}} - \frac{\phi_{q1}}{u_{q1}} - \frac{\phi_{23}}{u_{23}}\right]. \tag{82}$$

On the solution (79), Eq. (82) becomes

$$\mathcal{I}_{AB'} = 0, \tag{83}$$

*i.e.*, the balanced crossing PEE vanishes[6]. This is in contrast to the AdS$_3$ case, where the crossing PEE is a non-zero constant.

The vanishing of the crossing PEE may have a special physical meaning related to the Markov recovery process. This follows from the observation that the PEE, hence also the BPE and the crossing PEE, can be expressed in terms of the conditional mutual information (CMI) [63]. The CMI for a three-party system is defined as [73]

$$I(A : B | C) = S_{AC} + S_{BC} - S_{ABC} - S_C = I(A : BC) - I(A : C). \tag{84}$$

Here we are interested in the crossing PEE. One can check that the crossing PEE $\mathcal{I}_{AB'}$ can be written as a CMI

$$\begin{aligned}
\mathcal{I}_{AB'} &= \frac{1}{2}(S_{AA'} + S_{AB} - S_{A'} - S_B) \\
&= \frac{1}{2}(S_{BB'} + S_{AB} - S_{ABB'} - S_B) \\
&= \frac{1}{2}I(B' : A | B),
\end{aligned} \tag{85}$$

where in the second line we used the fact that $ABA'B'$ is in a pure state. More explicitly, the above equation (85) states that given a purification $ABA'B'$, the crossing PEE captures the correlation between $B'$ and $A$ from the point of view of $B'$. The CMI is also symmetric in $A$ and $B'$, and it can also be expressed as $I(B' : A | A')/2$.

Expressing the crossing PEE in terms of the CMI allows us to express it in terms of a specific Markov recovery process. To give a general overview of Markov recovery, consider a three-party system comprised of $A, B$ and $C$. Define the reduced density matrix of $A$ and $B$ as $\rho_{AB}$, which is generally a mixed state. We define a map $\mathcal{M}_{B \to BC} : B \to BC$, such that it acts on $\rho_{AB}$ and produces a tripartite state $\tilde{\rho}_{ABC}$, such that [73]

$$\tilde{\rho}_{ABC} = \mathcal{M}_{B \to BC}(\rho_{AB}). \tag{86}$$

The question we want to address is whether it is possible to perfectly recover a tripartite state $\rho_{ABC}$ under a mapping (86), provided the recovery mapping $\mathcal{M}_{B \to BC}$ exists. In fact, the recovery could be perfect or approximate (imperfect) [93]. The degree of imperfectness is given by the fidelity[7]

$$\max_{\mathcal{M}_{B \to BC}} \mathcal{F}(\rho_{ABC}, \mathcal{M}_{B \to BC}(\rho_{AB})) \geq e^{-I(A:C|B)}, \tag{87}$$

---

[6]Similarly, the difference between the entanglement negativity and half of the mutual information was found to be vanishing [83]. This is consistent with our results, since the entanglement negativity is also claimed to be the dual to the EWCS.

[7]The fidelity between two density matrices $\rho_1$ and $\rho_2$ is given by $\mathcal{F}(\rho_1, \rho_2) = \left(\text{Tr}\sqrt{\sqrt{\rho_1}\rho_2\sqrt{\rho_1}}\right)$.

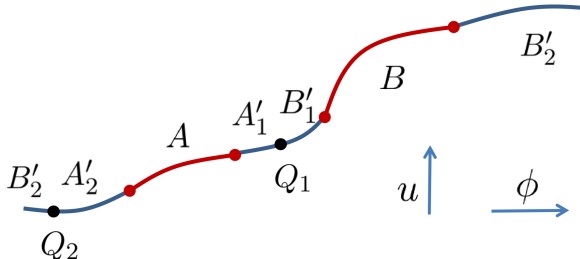

Figure 9: The non-adjacent intervals $A : \{(u_1, \phi_1), (u_2, \phi_2)\}$ and $B : \{(u_3, \phi_3), (u_4, \phi_4)\}$.

*i.e.*, it is lower bounded by the exponential of the (negative) CMI and ranges between $[0, 1]$, *i.e.*, $0 \leq \mathcal{F}(\rho_1, \rho_2) \leq 1$. If the two states are equal, then the fidelity is 1. On the other hand, the fidelity vanishes if the states are infinitely far away from each other in the Hilbert space, *i.e.*, if they are orthogonal to each other. The recovery is exact when the CMI vanishes, whereas for other cases, the recovery is approximate. In both cases, on of the goals is to find an explicit expression of the recovery map. Authors in [94,95] have, in fact, considered such maps, which is known as the Petz map [96,97].

When the partition of $A'B'$ satisfies the balance condition, the crossing PEE, as we have argued, is a generalization for the Markov gap. More interestingly, in 3d flat holography, the crossing PEE vanishes *i.e.,*

$$\mathcal{I}_{AB'}|_{\text{balance}} = \mathcal{I}_{BA'}|_{\text{balance}} = 0, \tag{88}$$

and hence, the following CMI vanish

$$I(A : B|B')|_{\text{balance}} = I(B : B'|A')|_{\text{balance}} = 0. \tag{89}$$

According to Eq.(87), we can write

$$0 \geq -\frac{1}{2} \max_{\mathcal{M}_{A \to AA'}} \log \mathcal{F}\big(\rho_{AA'B}, \mathcal{M}_{A \to AA'}(\rho_{AB})\big)\Big|_{\text{balance}}, \tag{90}$$

$$0 \geq -\frac{1}{2} \max_{\mathcal{M}_{B \to BB'}} \log \mathcal{F}\big(\rho_{ABB'}, \mathcal{M}_{B \to BB'}(\rho_{AB})\big)\Big|_{\text{balance}}. \tag{91}$$

Since the fidelity ranges from 0 to 1, the above inequalities can only be satisfied when the fidelity is 1. This implies that the vanishing crossing PEE (or Markov gap) implies a perfect Markov recovery from $\rho_{AB}$ to $\rho_{ABB'}$ or $\rho_{AA'B}$ where $A'$ and $B'$ are determined by the balance conditions.

## 5.3 BPE and EWCS for non-adjacent cases

Then we turn to the non-adjacent case where $A$ and $B$ are given by

$$A : \{(u_1, \phi_1), (u_2, \phi_2)\}, \qquad B : \{(u_3, \phi_3), (u_4, \phi_4)\}. \tag{92}$$

Here we set $u_4 > u_3 > u_2 > u_1$, hence keeping the degrees of freedom in $AB$ in some order. The complement $A'B'$ is then partitioned by two points

$$Q_i : (u_{q_i}, \phi_{q_i}), \qquad i = 1, 2. \tag{93}$$

See Fig. 9. The EWCS for this case was also given in [83], which is given by

$$E_W(A:B) = \frac{1}{4G_N} \left| \frac{X}{\sqrt{T}(1-T)} \right|, \tag{94}$$

where $T$ and $X$ are defined by

$$T = \frac{u_{12}u_{34}}{u_{13}u_{24}}, \qquad X = T\left(\frac{\phi_{12}}{u_{12}} + \frac{\phi_{34}}{u_{34}} - \frac{\phi_{13}}{u_{13}} - \frac{\phi_{24}}{u_{24}}\right). \tag{95}$$

Then let us calculate the BPE$(A:B)$. Again we consider the covariant configuration, and hence we need four parameters to determine the two partition points. However, the balance conditions (44) only give two equations

$$\frac{\phi_{2q_1}}{u_{2q_1}} - \frac{\phi_{q_13}}{u_{q_13}} = \frac{\phi_{q_22}}{u_{q_22}} - \frac{\phi_{q_23}}{u_{q_23}}, \qquad \frac{\phi_{q_21}}{u_{q_21}} - \frac{\phi_{q_24}}{u_{q_24}} = \frac{\phi_{q_11}}{u_{q_11}} - \frac{\phi_{q_14}}{u_{q_14}}, \tag{96}$$

which are not enough to determine the two partition points. For simplicity we can set

$$(u_1, \phi_1) = (0, 0), \tag{97}$$

due to the translation symmetries along $u$ and $\phi$. Solving the above equations we get $\phi_{q_1}$ and $\phi_{q_2}$ in terms of $\phi_i, u_i, u_{q_1}$ and $u_{q_2}$, where $i = 1, 2, 3, 4$. Then we are left with two undetermined parameters $u_{q_1}$ and $u_{q_2}$.

One may boldly expect that, as in the adjacent case, when we substitute the solutions for $\phi_{q_1}$ and $\phi_{q_2}$ into the PEE $s_{A_1'AA_2'}(A)$, we will get a result exactly equals to the EWCS (94), which is independent of $u_{q_1}$ and $u_{q_2}$. However in this case we obtain

$$\text{BPE}(A:B) = \frac{c_M u_{q_1 q_2}\left(u_3\left(u_4 u_{34}\phi_2 + u_2 u_{23}\phi_4\right) - u_2 u_4 u_{24}\phi_3\right)}{12 u_4 u_{23}\left(u_2 u_3\left(u_4 - u_{q_1}\right) + \left(\left(u_2 + u_3 - u_4\right)u_{q_1} - u_2 u_3\right)u_{q_2}\right)}, \tag{98}$$

which indeed depends on $u_{q_1}$ and $u_{q_2}$. If we take the limit $\phi_3 \to \phi_2$ first, and then take the limit $u_3 \to u_2$, we will find $(\phi_{q1}, u_{q1}) = (\phi_2, u_2)$ and the BPE (98) will reproduce the result for the adjacent case.

We then compute the difference between BPE$(A:B)$ (98) and $E_W(A:B)$ (94) and find that, as long as one of the following two conditions is satisfied [8]

$$u_{q_1} = \frac{u_2 u_4}{(1 - \sqrt{T})u_2 + \sqrt{T}u_4}, \tag{99}$$

$$u_{q_2} = \frac{u_2 u_4}{(1 + \sqrt{T})u_2 - \sqrt{T}u_4}, \tag{100}$$

then we will have

$$\text{BPE}(A:B) = E_W(A:B). \tag{101}$$

Note that the right-hand side of (99) (or (100)) only depends on the $u$ coordinates of the four endpoints of $AB$. This makes (99) and (100) quite a simple requirement as an additional constraint. In summary, when the two balance conditions together with one additional simple requirement ((99) or (100)) are satisfied, we will get a BPE that equals to the EWCS and is independent of the rest of one free parameter. This observation indicates that the BPE we

---

[8]We thank Debarshi Basu for pointing out a typo in the equations (99) and (100) in the first version of the paper.

defined with only two balance conditions is closely related to the EWCS. If they are not related, we will need more than one requirement to equate the BPE and EWCS since there are still two free parameters. Secondly, for covariant configurations, where we need twice the number of parameters to determine the partition points for $A'B'$, the definition of the BPE needs further constraints along with the balance conditions.

We may also regard the additional condition (99) (or (100)) as the balance requirement, a fact which we did not understand until recently. In a recent work [98], authors studied the BPE for the BMSFTs dual to topologically massive gravity in asymptotically flat spacetimes. In these configurations, we have $c_L \neq 0$, and hence the entanglement entropies for single intervals contain a contribution from the logarithmic term in (76). It turns out that the logarithmic term should also satisfy an independent balance requirement that coincides with (99) (or (100)). Thus, our claim that the BPE gives the EWCS for non-adjacent cases in 3-dimensional flat holography is perfectly justified.

## 5.4 Balanced crossing PEEs and tripartite entanglement

It is interesting to visualize the interpretation of the crossing PEE or the generalized Markov gap from the gravity side. For the adjacent intervals, we have explicitly shown that the balanced crossing PEEs can be regarded as the generalized Markov gap. The Markov gap is supposed to be lower bounded by the ratio of the AdS radius and Newton's constant multiplied by the EWCS endpoints as shown in [73]. Indeed, this is true for canonical purification, but it is unclear what happens for generic purifications. The universal nature of the balanced crossing PEE is expected to show some bound from the gravity side as well. Thus, for a generic purification, we propose the following[9]

$$\sum \text{crossing PEE}\Big|_{\text{balance}} \geq \frac{\log 2}{4\,G_N}|\partial\Sigma_{AB}|. \tag{102}$$

Here in the sum, one needs to include the contribution from all the crossing PEEs on the balanced condition, and $|\partial\Sigma_{AB}|$ accounts for the number of EWCS endpoints. From Eq.(55), we can evidently see that the purification from path-integral optimization actually saturates the above bound. The non-vanishing crossing PEEs suggest that the states might have large tripartite entanglement [36]. On the other hand, we see the crossing PEE vanishes for the BMSFT dual to Einstein gravity, which violates the above inequality (102). This could imply the following two different cases: either the BMSFT states have the bipartite or GHZ state-type entanglement (sometimes they are referred to as the sum of triangle states (SOTS)) [71,72], or the bound depends only on $c_L$, but not on $c_M$. In any case, Eq.(102) needs more investigation for both AdS and non-AdS gravity as well as for generic CFTs, and we hope to address them in the future.

# 6 Summary and outlook

The primary objective of this paper is to demonstrate BPE as a proper measure for the total intrinsic correlations in a mixed state. First, we argued that the BPE is purification-independent by showing that, given a mixed state the BPE is the same for different purifications, including the class we constructed from the Euclidean path-integral, the canonical purification and pure states with a gravity dual. Secondly, in all the configurations where the BPE can be evaluated,

---

[9]We have not explicitly found a similar statement for the non-adjacent case. This is because, the way we have defined, the crossing PEEs do not reduce to the Markov gap. However, it would be interesting to see whether, for the non-adjacent case, some modified version of the crossing PEE respects a similar bound.

the reflected entropy for the canonical purification turns out to be a specific case of the BPE. This implies that BPE holds for any purifications and generalizes the reflected entropy. Finally, we find that the correspondence between the BPE and the EWCS goes beyond AdS/CFT. We conducted a detailed evaluation of the BPE in 3d flat holography, coinciding with the EWCS result.

The purification independence for BPE was not addressed in [1] because of the two special cases of purifications which give different BPE and balanced crossing PEEs from other purifications. The first one is the pure state constructed using the so-called state-surface correspondence [99] where the pure state is settled at the union of the boundary interval $AB$ and its minimal surface $\mathcal{E}_{AB}$. In this case the BPE$(A:B)$ differs from the EWCS and the balanced crossing PEE $\mathcal{I}_{AB'} = c/12\log 2$ (see section 3.3 in [1] for details). The second case is the pure state calculated by the optimized path-integral, which is claimed to be the minimal purification where $S_{AA'}$ gives the EoP, as well as the EWCS [70]. However, both of the two cases are not robust enough to exclude the purification independence of the BPE. Especially, the support for the surface-state correspondence is far from enough; hence the pure states constructed under this context may not exist. Also, the existence of negative mutual information in the optimized purification is subtle.

We also find that the minimized crossing PEE is a natural generalization for the Markov gap. On the other hand, our crossing PEE construction covers more general cases and goes beyond the canonical purification. We decompose the BPE into two parts, the intrinsic PEE $\mathcal{I}_{AB}$ and the crossing PEEs. More interestingly, the crossing PEE (or their sum) at the balance point is shown to be minimal. For the adjacent cases, the minimized (or balanced) crossing PEE is the generalized version of the Markov gap, and it is observed to be universal, which is determined by the central charge alone. The minimized crossing PEE may capture a universal aspect for the entanglement structure in quantum systems. It may play an essential role in quantum information. One example we discussed is that, since the balanced crossing PEE can be expressed as a CMI, it can be used to characterize how precisely the Markov recovery process can be conducted. Remarkably, in the BMSFT that duals to 3$d$ flat space in Einstein gravity, the balanced crossing PEE vanishes, suggesting the possibility of a perfect Markov recovery process. Furthermore, we interpret the crossing PEEs as a signature of tripartite entanglement.

**Entanglement of purification revisited**

If the minimized crossing PEE is purification-independent, then the EoP may be explicitly calculated in the context of PEEs. As we showed that the crossing PEE at the balance point is minimized, the minimized $S_{AA'}$ among all the purifications and partitions should be

$$S_{AA'}|_{\min} = \mathcal{I}_{AB} + (\mathcal{I}_{AB'} + \mathcal{I}_{A'B})|_{\text{minimized}} + \mathcal{I}_{A'B'}|_{\text{minimized}}. \tag{103}$$

In the adjacent case, on the right-hand side, the first term is independent of the purifications. The second term is evaluated at the balanced point and is given by a universal constant. The third term is purification dependent and could be turned off for some special purifications. Hence we conclude that

$$E_p(A:B) = S_{AA'}|_{\min} = \frac{1}{2}I(A:B) + \frac{c}{3}\log 2, \tag{104}$$

which is greater than the EWCS by a constant $c/6\log 2$. This result apparently contradicts the claim of [70] that the EoP gives the EWCS. The contradiction lies within the exclusion of the negative PEE $\mathcal{I}_{A'B'} = -c/6\log 2$. After including this term, the conjecture perfectly holds. The negative PEE can be fixed by a constant shift for the scalar field; hence the negative PEE in (42) can be shifted to zero. However, under this shift, the $S_{AA'}$ also changes to be (104).

**Future directions**

Though the concept of BPE and the crossing PEEs are inspired by our study for holographic systems, they can be defined on generic quantum systems. Testing the purification independence for the BPE and universality of the balanced crossing PEE in more generic configurations will be some important future directions.

- One can consider more generic purifications in condensed matter systems. In our paper, we mainly focus on the vacuum states, and it will be crucial to test the purification independence for the BPE in other pure states which are excited. For example, it would be interesting to compute BPE in 2d free CFTs explicitly and compare it with existing techniques [100] to see the numerical advantages.

- Both the EWCS and reflected entropy can be defined in higher dimensions, and it will be interesting to test the correspondence between the EWCS and the BPE in higher dimensions. In some highly symmetric configurations, the ALC proposal can still be valid. One can also use the formula for the so-called extensive mutual information [28, 101] to compute the PEE in higher dimensional CFTs.

- One workable case is the (warped) $AdS_3$/ warped CFT correspondence [102,103], where the geometric picture for entanglement entropy was also worked out in [55,82,104]. In this case, EWCS can be constructed in a similar way as in the flat case; hence it is also possible to test the correspondence between the EWCS and the BPE.

- Our calculations of BPE can also be generalized to the finite temperature cases and for other gravity duals like topological massive gravity (TMG) with non-zero $c_L$. The crossing PEE is supposed to be non-zero and should depend on the topological term as obtained in the context of the entanglement negativity [83]. The generalization of the EoP and EWCS from bipartite states to multipartite states was explored in [105, 106], it will also be interesting to explore the similar generalization for the BPE.

- It will be interesting to explore the dynamics of the BPE and, more generally, the crossing PEEs by inserting heavy operators, which can be understood as the shock-wave perturbation from the dual geometry [107, 108].

- Both the BPE and the entanglement negativity are measures of mixed state correlations, which in holographic CFTs are captured by the same dual, *i.e.*, the EWCS [17]. However, the entanglement negativity is computed directly via the density matrix, while a definition for the PEE or BPE based on the density matrix is still not clear. Hence a direct comparison between them is quite tricky. Nevertheless, exploring the relation between the BPE and entanglement negativity is an interesting avenue of research.

- The entanglement negativity contour was previously examined in [54], where a version similar to the ALC proposal for negativity was introduced. It will be possible to impose balance conditions on the negativity, which might also provide a version of BPE for the negativity.

So far, quantities like the reflected entropy, EWCS, and EoP in covariant configurations are rarely studied (see [109, 110] for examples). The BPE we have defined can be naturally extended to covariant configurations. Our calculation in 3-dimensional flat holography shows a perfect match between the BPE and the EWCS in totally covariant configurations. It will be very interesting to explore the covariant configurations in AdS/CFT. In the static configurations where the subsystems and the complement $A'B'$ are confined on a time slice, the position for any partition point in $A'B'$ is determined by a single parameter and the number of the balance

conditions equals the number the partition points. This helps us determine all the positions of the partition points using the balance requirements. However, for covariant configurations, one needs two parameters to determine the position of one partition point, while the number of balance conditions is the same as the static case, which is not enough to determine the positions of all the partition points in $A'B'$. Similar to [98], we expect additional balance requirements to appear if we consider the more generic CFTs with different left and right moving central charges. We hope to come back to this point in the near future.

## Acknowledgements

We wish to thank Pawel Caputa, Ling-Yan Hung, Jonah Kudler-Flam, Masamichi Miyaji, Huajia Wang for discussions and Pawel Caputa, Juan F. Pedraza, Tadashi Takayanagi for comments on the draft. We thank the anonymous referees of SciPost Physics for helpful suggestions. HC is partially supported by the International Max Planck Research School for Mathematical and Physical Aspects of Gravitation, Cosmology and Quantum Field Theory and by the Gravity, Quantum Fields and Information (GQFI) group at the Max Planck Institute for Gravitational Physics (Albert Einstein Institute). The GQFI group is supported by the Alexander von Humboldt Foundation and the Federal Ministry for Education and Research through the Sofja Kovalevskaja Award. PN is supported by University Grants Commission (UGC), Government of India. QW and HZ are supported by the "Zhishan" Scholars Programs of Southeast University.

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
