# Peer review of "Balanced Partial Entanglement and Mixed State Correlations"

_SciPost Physics, doi:SciPost Phys. 12, 137 (2022)_

## Round 1 · Referee Report · Anonymous (Referee 1) · 2022-3-2

Strengths

  1. The authors propose a new quantity as the measure of the intrinsic correlation between two subsystems in a mixed state.

  2. There are several positive examples supporting the proposal.

Weaknesses

  1. There is an example of puzzling negative partial entanglement entropy and mutual information without a good explanation.

  2. It is still an open question whether the proposed balanced partial entanglement entropy is purification-independent.

Report

The authors study a special kind of the partial entanglement entropy (PEE), the balanced partial entanglement entropy (BPEE), as a measure of the intrinsic correlation between two subsystems $A$ and $B$ in a mixed state. For a fixed auxiliary system $A’\cup B’$ that purifies $A\cup B$, the authors vary the relative sizes of $A’$ and $B’$ and define the PEE under some balance condition as BPEE. It is argued that the BPEE is independent from the purification, and always equals the reflected entropy, which is defined in the canonical purification. It is also argued that in holographic theories the BPEE is dual to the entanglement wedge cross section (EWCS), as the reflected entropy is believed to be dual to the EWCS. Several examples are given to support the claim of in this paper, while there also exist examples that contradict the claim.

Requested changes

Though whether the BPEE is purification-dependent is an open question, I think the proposal and checks in this paper are interesting and meaningful and the paper deserves publication. I have several comments for the authors, while it is up to the authors whether to follow these comments.

1. In the purification the authors consider, it is not required that the Hilbert space of $A’\cup B’$ has the same dimension as the Hilbert space of $A\cup B$. It is also not required that the Hilbert space of $A’$ has the same dimension as the Hilbert space of $A$ or the Hilbert space of $B’$ has the same dimension as the Hilbert space of $B$. There is another way to impose the balance condition. One could consider the purifications $A’\cup B’$ with the condition that the Hilbert space of $A’$ has the same dimension as the Hilbert space of $A$ and the Hilbert space of $B’$ has the same dimension as the Hilbert space of $B$. There are an infinite number of such purifications, and one could choose the purification with the balance condition satisfied and define the PEE in such a purification as the BPEE. If needed, the minimal condition could also be imposed. The BPEE defined in this way is apparently intrinsic. If this definition of BPEE has any relation with the BPEE defined in this paper?

2. As the authors have stated for the negative PEE and mutual information, it is indeed puzzling. The Araki-Lieb inequality should hold for any reasonable quantum state. Could the authors comment more about the strange result? Is it because of the pathology of the model, the pathology of the state, or the problem of the calculation method?

3. When mentioning the entanglement negativity, the authors may consider mentioning the first works calculating the entanglement negativity in quantum field theory 1206.3092 and 1210.5359.

4. There are several small grammatical errors. For example, above eq (71) on page 18, there are “… can be trace back to …”, “… does not reduces to …”, “… one needs to more careful about …”. I suggest the authors check and correct similar errors in the whole paper.

  • validity: high
  • significance: high
  • originality: high
  • clarity: high
  • formatting: excellent
  • grammar: good

Author:  Qiang Wen  on 2022-03-23  [id 2317]

(in reply to Report 1 on 2022-03-02)

Though whether the BPEE is purification-dependent is an open question, I think the proposal and checks in this paper are interesting and meaningful and the paper deserves publication. I have several comments for the authors, while it is up to the authors whether to follow these comments.

Reply: The authors thank the Referee very much for the recommendation.

  1. In the purification the authors consider, it is not required that the Hilbert space of A′∪B′ has the same dimension as the Hilbert space of A∪B. It is also not required that the Hilbert space of A′ has the same dimension as the Hilbert space of A or the Hilbert space of B′ has the same dimension as the Hilbert space of B. There is another way to impose the balance condition. One could consider the purifications A′∪B′ with the condition that the Hilbert space of A′ has the same dimension as the Hilbert space of A and the Hilbert space of B′ has the same dimension as the Hilbert space of B. There are an infinite number of such purifications, and one could choose the purification with the balance condition satisfied and define the PEE in such a purification as the BPEE. If needed, the minimal condition could also be imposed. The BPEE defined in this way is apparently intrinsic. If this definition of BPEE has any relation with the BPEE defined in this paper?

Reply: Thanks for the suggestion. We think the biggest disadvantage of the EoP and the reflected entropy is their restriction to certain specific purification. This makes them automatically intrinsic, but also hard to evaluate (especially for the EoP) and hard to investigate other properties like monotonic under inclusion. However, the information (density matrix) of the mixed state is entirely included in any purification, hence any intrinsic correlation in the mixed state could be somehow extracted in any purification. On the other hand, if someone gives a proposal that works for general purifications, then it comes with the problem of demonstrating its intrinsic nature or the independence from purification.

The referee suggested a way to demonstrate the purification-independence of the BPE. If we require that A and A’ (B and B’) to have Hilbert space of the same dimensions, and choose those that satisfy the balance conditions, this should give us the BPE we have defined in this special class of purifications. According to our claim that the BPE is purification independent, the BPE in these configurations should all equal to the reflected entropy. One could arrive at such configurations by starting from the canonical purification and applying local unitary transformations in A’ and B’ respectively. The dimension of the Hilbert space of A’ and B’, as well as the BPE, are invariant under such local unitary transformations. However, the right way to demonstrate that the intrinsic nature of BPE is to go beyond this class of purifications. We can test this property in more generic purifications, while a general proof is not clear to us.

Let us also introduce some related facts about the EoP. In exploratory numerical computations done for EoP using Gaussian techniques for lattice discretizations of free QFTs (SciPost Phys. 10 (2021) 3, 066), it was found that the value of the EoP is independent of the dimension of the purifying Hilbert space. This statement was verified for the most general Gaussian purifications of Gaussian mixed states defined in free bosonic and fermionic (1+1)-dimensional CFTs (e.g. free boson and critical transverse Ising). However, it is currently not known whether this result also applies to more general (i.e. non-Gaussian) purifications of Gaussian mixed states, or even if this holds for more general mixed states. It is an open problem to understand this situation for EoP in quantum many-body systems. We believe the situation for BPE should in principle be similar: i.e. invariant under the changing of the dimension of the purifying Hilbert space. Of course, this should be verified numerically in free QFTs, specially since there could exist a large class of purifications which satisfy the balance condition.

  1. As the authors have stated for the negative PEE and mutual information, it is indeed puzzling. The Araki-Lieb inequality should hold for any reasonable quantum state. Could the authors comment more about the strange result? Is it because of the pathology of the model, the pathology of the state, or the problem of the calculation method?

Reply: We admit that the negative PEE and MI in the vacuum state prepared by the optimized path integral looks puzzling.

One possible reason that causes the puzzle may come from the finite cutoff after optimization. As we can see, in the optimized path integral the cutoff scale becomes finite and is comparable to the length of the sub intervals like A, B, B’. When the cutoff scale excels some critical value, the formula (29) to evaluate entanglement entropy may become invalid.

Another possible explanation for this puzzling observation may be given by looking at the gate-counting perspective of path-integral optimization and the quantum state obtained thereof. In (1904.02713) authors studied whether it was possible to recover the Liouville action as a complexity cost functional for states constructed from Euclidean-time evolution in 2d CFTs from a quantum circuit perspective. Indeed this is possible by the inclusion of non-unitary gates in the quantum circuit. As a consequence, the inclusion of said non-unitary operations on the circuit acting on the initial (2d CFT ground) state may be responsible for yielding a final state (obtained by the optimization of the Euclidean path-integral) which violates the inequality.

Also, one of the authors in [70] suggested an explanation for the puzzle. The equation of motion of the Liouville action allows a constant shift for the scalar field. After a proper choice for the constant, the negative PEE or MI can become zero or positive. Nevertheless, under this shift, the entanglement entropy $S_{AA’}$ also changes and differ from the EWCS by a constant of c/6 log2.

Indeed this puzzle help us get rid of one counter-example that contradicts with the purification independence of the BPE. If the following two statements are true at the same time: 1, in the vacuum state from optimized path-integral $S_{AA’}=EWCS/4G$; 2, the PEE (or MI) cannot be negative; then the BPE in this state cannot capture the EWCS because it is smaller than $S_{AA’}$. Fortunately, the puzzle tells us that the above two statement cannot be true at the same time.

  1. When mentioning the entanglement negativity, the authors may consider mentioning the first works calculating the entanglement negativity in quantum field theory 1206.3092 and 1210.5359.

Reply: Thanks for pointing out the two references, we will add them in the next version.

  1. There are several small grammatical errors. For example, above eq (71) on page 18, there are “… can be trace back to …”, “… does not reduces to …”, “… one needs to more careful about …”. I suggest the authors check and correct similar errors in the whole paper.

Reply: Thanks, we will fix these typos, and further proofread the manuscript.

---

## Round 1 · Referee Report · Anonymous (Referee 2) · 2022-3-20

Strengths

Proposal of a measure of intrinsic quantum correlations which exhibits universal properties.

Weaknesses

No connection with an existing measure of quantum correlations in mixed states, the entanglement negativity.

Report

In this paper, the authors analyse a measure of the intrinsic correlations between two subsystems in a mixed state, the balanced partial entanglement (BPE) entropy. In particular, they prove that this quantity is independent from the purifications based on the Euclidean path-integral, even though it is still unclear if this holds for any kind of purification. They study the connection between the BPE and the reflected entropy and this allows the authors to define a generalisation of the Markov gap from the canonical purification to more generic purifications. Finally, they study the dual gravity side of the BPE, which is the entanglement wedge cross-section (EWCS).
The paper is well-written and it contains some nontrivial results on a recently popular topic. The calculations are clearly explained and detailed. Moreover, the authors compare their results with new information-theoretic quantities, like the reflected entropy or the entanglement of purification. Therefore, I recommend the paper for publication.

Requested changes

Here is nevertheless a short list of comments/questions/typos: - Another relevant measure of quantum correlations in mixed states is the negativity. Could the authors clarify the difference between the two entanglement measures? There are proposals that claim that the EWCS is dual to the logarithmic negativity (see e.g. Phys. Rev. D 99 (2019) 106014): could the authors relate this to the duality between EWCS and BPE? - What is the relation between the BPE and other entanglement measures, like entanglement negativity, in non-holographic systems? - Before and in Eq. 10, why the authors use $S_{AA'}(A)$ rather than $s_{AA'}(A)$? Is this a typo? - Have the authors thought if a quantity similar to the partial entanglement entropy could be defined from the negativity contour rather than from the entanglement one? - Some typos could be: pag. 4 (before Eq. 6) each degrees $\to$ each degree, pag. 25 einstein gravity $\to$ Einstein gravity, pag. 26 tripartire $\to$ tripartite.

  • validity: good
  • significance: good
  • originality: good
  • clarity: good
  • formatting: excellent
  • grammar: excellent

Author:  Qiang Wen  on 2022-03-23  [id 2316]

(in reply to Report 2 on 2022-03-20)

The authors thank the referee very much for comments and recommendation.

  1. Another relevant measure of quantum correlations in mixed states is the negativity. Could the authors clarify the difference between the two entanglement measures? There are proposals that claim that the EWCS is dual to the logarithmic negativity (see e.g. Phys. Rev. D 99 (2019) 106014): could the authors relate this to the duality between EWCS and BPE?

Reply: Both the BPE and the negativity are measures of mixed state correlations which in holographic CFTs are captured by the same dual: the entanglement wedge cross section (EWCS). However, their field-theoretic definitions are very distinct and may therefore be seen a priori as independent correlation measures. The most evident difference between these quantities is the fact that while the negativity is defined intrinsically for mixed states via the partial transpose of the density matrix, the BPE is defined via certain purifications of the mixed state which satisfies the balance condition, and calculated by certain linear combination of subset entanglement entropies. Of course, the striking aspect that these quantities, alongside others, have been conjectured (or shown) to have the same gravitational dual in holographic CFTs points to the fact that they share a certain amount of information about the intrinsic correlations of mixed states which is highlighted in the holographic cases.

It was demonstrated in [17] that both of the logarithmic negativity and the Renyi reflected entropy can be calculated by the correlation functions of twist operators, and the results match with each other. The coincidence between the negativity and the reflected entropy furthermore lead to the duality between the negativity and the EWCS. Although, the BPE is also closely related to the reflected entropy, its definition based on the density matrix of the mixed state is still not established. So far, the way we calculate the PEE or BPE is either the ALC proposal (linear combination of subset entanglement entropies) or the extensive mutual information proposal (directly solving the 7 requirements for PEE for certain theories), which are very different from the way using twister operators. Hence, a direct comparison between the BPE and negativity is hard to conduct currently. We hope to come back to this point in the future. We added this as a bullet point of section 6.

  1. What is the relation between the BPE and other entanglement measures, like entanglement negativity, in non-holographic systems?

Reply: In non-holographic systems, the reflected entropy is also a special case of the BPE defined on the canonical purification. Also, our discussion on the comparison between the BPE and EoP extend to the non-holographic cases. As we mentioned previously, the relations between BPE and other entanglement measures, such as odd entropy and logarithmic negativity in general CFTs or QFTs are currently unknown. In the set-up of thermal states or subsystems of ground states of non-holographic free QFTs, these quantities can efficiently computed using numerical method, which lead to a tractable comparison between these quantities. We hope to return to this question in future work.

  1. Before and in Eq. 10, why the authors use $S_{AA’}(A)$ rather than $s_{AA’}(A)$? Is this a typo?

Reply: Yes, the referee is right. Thanks for pointing it out.

  1. Have the authors thought if a quantity similar to the partial entanglement entropy could be defined from the negativity contour rather than from the entanglement one?

Reply: Entanglement negativity contour was previously examined in [54] where a version similar to the ALC proposal for negativity was introduced. It will be possible to impose balance conditions on the negativity, which might also provide a version of BPE for the negativity and possibly dual to the EWCS. We have added this comment to the main text as the final bullet point of section 6.

  1. Reply: We have fixed all the typos pointed out by the referee. Thanks.

---

## Round 2 · Referee Report · Anonymous (Referee 2) · 2022-3-23

Report

I am happy with the answers/clarifications made by the authors and I recommend this article for publication.

---

## Round 2 · Referee Report · Anonymous (Referee 1) · 2022-4-6

Report

I thank the authors for the replies to my comments and questions. I am happy to recommend the paper for publication in its present form.

---

## Round 2 · List of Changes

1. we fixed the typos pointed out by the referees and did further proofreading

  2. we added a paragraph above section 5.4 in page 25, to introduce a follow up work that solves our confusion about the additional balance requirement.

  3. we added two bullets in the discussion section related to the entanglement negativity.

  4. we added four references.

  5. the paragraphs below the bullets in section 6 is reformulated.

---

## Editorial Decision

published